# Influence of multidecadal variability on high and low flows: the case of the Seine basin

Rémy Bonnet[1], Julien Boé[1], and Florence Habets[2,3]

[1]CECI, Université de Toulouse, CNRS, Cerfacs, Toulouse, France
[2]UMR 8538, Laboratoire de Géologie de l'Ecole Normale Supérieure, CNRS, PSL
[3]UMR 7619 METIS, Sorbonne Université-Faculté des Sciences, CNRS

**Correspondence:** Bonnet (remy.bonnet@ipsl.fr)

**Abstract.** The multidecadal hydroclimate variations of the Seine basin since the 1850s are investigated. Given the scarcity of long term hydrological observations, a hydrometeorological reconstruction is developed based on hydrological modelling and a method that combines the results of a downscaled long-term atmospheric reanalysis and local observations of precipitation and temperature. This method improves previous attempts and provides a realistic representation of daily and monthly river flows. This new hydrometeorological reconstruction, available over more than 150 years while maintaining fine spatial and temporal resolutions, provides a tool to improve our understanding of the multidecadal hydrological variability in the Seine basin, as well as its influence on high and low flows. This long term reconstruction allows analysis of the strong multidecadal variations of the Seine river flows. The main hydrological mechanisms at the origin of these variations are highlighted. Spring precipitation plays a central role by directly influencing the multidecadal variability in spring flows, but also soil moisture and groundwater recharge, which then regulate summer river flows. These multidecadal hydroclimate variations in the Seine basin are driven by anomalies in large scale atmospheric circulation, which themselves appear to be influenced by sea surface temperature anomalies over of the North Atlantic and the North Pacific. The multidecadal hydroclimate variations also influence high and low flows over the last 150 years. The analysis of two particularly severe historical droughts, the 1921 and the 1949 events, illustrates how long-term hydroclimate variations may impact short-term drought events, particularly through groundwater-river exchanges. The multidecadal hydroclimate variations described in this study, probably of internal origin, could play an important role in the evolution of water resources in the Seine basin in the coming decades. It is therefore essential to take the associated uncertainties into account in future projections.

## 1 Introduction

Future impacts of climate change on the hydrological cycle could have major socio-economic consequences (Pachauri et al., 2014). In France, important hydrological changes are expected at the end of the 21st century in response to global warming, with for example a strong general reduction of river flows in summer (Dayon et al., 2018). To develop robust adaptation strategies, an accurate assessment of the uncertainties in impact projections is crucial.

Internal climate variability is a major source of uncertainties in future projections, especially over the coming decades (Deser et al., 2012) and for precipitation (Hawkins and Sutton, 2009; Hingray and Saïd, 2014; Terray and Boé, 2013). Efforts to

better evaluate the capacity of climate and/or hydrological models to correctly reproduce hydroclimate variability are therefore necessary, to ensure that the uncertainties in impact projections are correctly estimated. As a first and mandatory step, it is necessary to better characterize and understand hydrological variations in the observations.

A large body of work has dealt with river flow variability over Europe. At interannual time scales, the large scale atmospheric circulation over the North-Atlantic plays a major role in hydrological variations over Europe (Kingston et al., 2006b, a; Bouwer et al., 2008; Steirou et al., 2017). The North Atlantic Oscillation (Cassou et al., 2004; Hurrell and Deser, 2009) in particular is known to influence river flows over Europe, mainly in winter (Kingston et al., 2006b, a; Bouwer et al., 2008; Steirou et al., 2017).

Regarding the uncertainties in impact projections due to internal variability, variations at longer time-scales, i.e. at decadal and multidecadal time-scales, are more important than interannual variations, as they may modulate the hydroclimate state over several decades, i.e. at climate time scales. They may temporarily reinforce or reduce, or even reverse the long-term impacts of climate change. Sutton and Dong (2012) have shown that over the 20th century, the Atlantic Multidecadal Variability (AMV; Schlesinger and Ramankutty, 1994; Kerr, 2000; Deser et al., 2010; Cassou et al., 2018), the main mode of decadal to multidecadal variability of the North Atlantic, characterized by basin-wide variations of sea surface temperatures (SST), impacts the European climate through changes in large scale atmospheric circulation. The positive phases of the AMV tend to reduce precipitation in spring over France. Recent studies have highlighted the existence of strong multidecadal variations in river flows over France (Boé and Habets, 2014; Bonnet et al., 2017). These variations, which can reach up to 40% of the long-term average, are especially important in spring and are consistent with the negative anomalies in spring precipitation associated with the AMV noted by Sutton and Dong (2012). The river flow variations over France however seem to lag the AMV (Boé and Habets, 2014). Consistently, large multidecadal variations have also been observed on a glacier in the French Pyrénées and are likely associated with the AMV (Marti et al., 2014). These multidecadal hydrological variations seem to modulate secular trends in river flows (Dieppois et al., 2016).

Additionally, the AMV could influence extreme hydrological events. Willems (2013) notes the existence of multidecadal variations in extreme precipitation and river flows over Europe, including France, attributed to changes in atmospheric circulation likely driven by the AMV. A recent study of Hodgkins et al. (2017) shows a significant positive influence of the AMV on floods over Europe. A positive relation has also been found between the AMV and heavy precipitation over Europe in all seasons (Casanueva Vicente et al., 2014). The role of warm Atlantic sea surface temperature as provider of moisture has been put forward to explain this link. Based on current literature, a competition may therefore exist between the AMV-driven changes in large scale circulation, that may lead to increases or decreases in precipitation and / or extreme precipitation over Europe, depending on the region and the season, and the increase in atmospheric moisture associated with the warmer SSTs of positive AMV phases. Regarding droughts, Giuntoli et al. (2013) show a statistical link between their severity for some French rivers and North Atlantic SSTs during the second half of the 20th century, which are both impacted by internal variability and anthropogenic forcings.

Climate and hydrological observations before the 20th century are rare. As a consequence, the studies discussed previously are very often limited to the 20th century or less, which represents a very short period to characterize decadal and multi-

decadal variability. The sampling uncertainties are indeed very large when statistics of multidecadal variations are calculated over barely more than 100 years and non-stationnarity may exist (Qasmi et al., 2017). Additionally, the climate of the 20th century, especially after 1950, is strongly influenced by external forcings (Bindoff et al., 2013), such as greenhouse gases or anthropogenic aerosols, which may also influence the hydrological cycle over Europe (e.g Boé, 2016). It is therefore difficult

to disentangle the respective roles of external forcings and internal variability in the decadal to multidecadal climate variations of the 20th century, and consequently to understand the responsible mechanisms. The shortness of the observational record has been recognized as a major challenge, as has the need to go further back in the past using innovative new approaches (e.g Cassou et al., 2018).

     Long-term observations of river flows are rare, with very few stations starting before 1900 for example, and they may

suffer from non-climatic anthropogenic influences as well as temporal inhomogeneities. Importantly, the other variables of the continental hydrological cycle, such as evapotranspiration, soil moisture or snow, are virtually not observed on such long time scales, which makes very difficult to understand the hydrological mechanisms at play.

     To move forward, long-term hydrometeorological reconstructions based on hydrological modelling have been developed (e.g Kuentz et al. 2015; Caillouet et al. 2016). Due to the scarcity of meteorological observations in the early 20th century

(Minvielle et al., 2015), the meteorological forcing needed for hydrological modelling must first be reconstructed. The recent release of long-term global atmospheric reanalyses (e.g. Twentieth Century Reanalysis (20CR, Compo et al. 2011) from the National Oceanic and Atmospheric Administration (NOAA)) opens great opportunities in that context. Statistical downscaling methods, typically used in climate change impact studies, can be applied to derive the high resolution meteorological forcing necessary for hydrological modelling from these global atmospheric reanalyses, as in Caillouet et al. (2016). This approach

presents two main limitations. First, the quality of the reconstruction depends on the quality of the reanalyses. As the density of assimilated observations (e.g. surface pressure in NOAA 20CR, Compo et al. 2011) strongly evolves over time, potential unrealistic trends and/or low frequency variations may exist (Krueger et al., 2013; Oliver, 2016; Bonnet et al., 2017). Second, this approach does not take advantage of the long-term local meteorological observations that may exist.

     Given these limitations, following the same general idea as Kuentz et al. (2015), Bonnet et al. (2017) presented a new

hybrid method that combines available long-term monthly observations of precipitation and temperature with the results of a statistical downscaling method applied to long-term atmospheric reanalyses. Compared to standard dynamical or statistical downscaling methods that only use large scale information and do not take advantage of local observations (e.g. temperature and precipitation) a more realistic representation of local hydroclimate variations can be obtained (Bonnet et al., 2017).

     In this study, we build upon the methodology developed in Bonnet et al. (2017) and target two main improvements. First, the

reconstruction is extended back in the past in order to assess more robustly the multidecadal variations. Given data availability, the reconstruction of Bonnet et al. (2017) begins in 1900. Focusing on the Seine basin (Figure 1), one of the main French river basins, we are able to extend the reconstruction back to the 1850s. The interest of this basin is the existence of long observational series of different variables contributing to the hydrological cycle, which are useful for developing and evaluating the reconstructions. Additionally, we improve the method to better capture daily variability, in order to have a finer reconstruction

of high and low flows. This allows to study whether high and low flows are influenced by multidecadal variations, which may have important socio-economic impacts.

This study has three main objectives: (i) to characterize the hydroclimate variations of the Seine basin since the 1850s, (ii) to study the hydrological and climatic mechanisms that cause these variations and (iii) to investigate the influence of these multidecadal variations on low and high flows.

The data, models and methods used are presented in section 2. The development of the hydrometeorological reconstruction is described in section 3. The hydrometeorological reconstruction is then evaluated against observations in section 4. From this reconstruction, the hydroclimate variations of the Seine basin are characterized in section 5, and the hydrological mechanisms associated with these variations are analysed. In section 6, the climate mechanisms underlying the hydrological multidecadal variations over the Seine basin are analysed. The link between these multidecadal variations and high and low flows is then explored in section 7. Two historical droughts are then analysed in detail. Finally, the main limitations of this study are discussed in section 8 and the conclusions and perspectives are drawn in section 9.

## 2 Data, models and methods

### 2.1 Observations

Daily river flows at 136 gauging stations over the Seine basin (Figure 1) from the national HYDRO database (www.hydro.eaufrance.fr) are used for the evaluation of the reconstruction. These river flow series, of variable lengths, all start before 1970 and may contain missing values. Some of the stations are very likely influenced by human activities (e.g. dams or water abstraction). These series are not homogenized, and therefore not necessarily free of measurement artifacts. The interpretation of observed variations at any single station must therefore be carried out carefully. Two long-term series, starting before 1900, are also available and used for the analysis of the link between extreme events and multidecadal hydroclimate variations, one at Paris Austerlitz on the Seine river (catchment area: $43800 km^2$) and one at Aisy-sur-Armançon on the Armançon river (catchment area: $1350 km^2$) (Figure 1). A long series of piezometric levels is also used to evaluate the hydrological reconstruction (Figure 1). The Beauce groundwater table at Toury, which has been monitored from the 1870s, is one of the few long piezometric measuring stations in France (Nicolas et al., 2013). These observations, which are used in particular to evaluate the hydrometeorological reconstruction developed in this study, are independent from it.

Different meteorological observations are used for the reconstruction. Monthly homogenized series of temperature and precipitation over the Seine basin from the Série Mensuelle de Référence (SMR, monthly series of reference) data set, developed by Météo-France (Moisselin et al., 2002) are used. 17 stations are available for precipitation and 4 for temperature, from 1885 to 2005. A long series of monthly precipitation at Paris (Slonosky, 2002), available since the late 17th century, is also used for the periods not covered by the SMR observations series at Paris, from 1852 to 1885 and from 2005 to 2008. Daily temperature and precipitation series from the Série Quotidienne de Référence (SQR, daily series of reference) data set between 1885-2003 are also used (Moisselin and Dubuisson, 2006). They are not homogenized. The number of stations varies greatly in time, from two to around sixty for precipitation and from one to seven for temperature.

## 2.2 The Safran-Surfex-AquiFR hydrometeorological system

The hydrological reconstruction is based on the French AquiFR hydrogeological modelling platform (Vergnes et al., 2019). AquiFR couples several hydrogeological models over France to the Surfex land surface model (Masson et al., 2013). The ISBA (Interactions between Soil, Biosphere, and Atmosphere, Noilhan and Planton, 1989) component of the Surfex modular land surface model, used for example in the French CNRM-CM5 coupled climate model (Voldoire et al., 2013), computes the water and energy exchanges at the interface between the soil, the vegetation and the atmosphere. In this study, version 8.1 of Surfex is used, with the multilayer version of ISBA (Decharme et al., 2013). The surface runoff and the drainage simulated by Surfex are then routed through the multi-layer aquifers and rivers of the Seine basin by AquiFR.

Over the Seine basin, the EauDyssée hydrogeological model (Saleh et al., 2011) is used in the AquiFR plateform. This hydrogeological model is an improved version of the Modcou hydrogeological model used in Bonnet et al. (2017). Unlike Modcou, the EauDyssée hydrogeological model implemented in AquiFR includes the simulation of river water levels, allowing for a better estimation of water exchanges between groundwater and rivers (Saleh et al., 2011). The exchanges between groundwater and rivers are computed daily based on the gradient head between the aquifer and the river, and a transfer coefficient that accounts for the river bed characteristic (see details in Vergnes and Habets, 2018). The water transfers in the unsaturated zone are improved (Philippe et al., 2011), which may be important considering that the unsaturated zone can reach 60 meters in the Seine basin, implying a delay in the groundwater recharge. Finally, AquiFR uses a 6-layer representation of the aquifers (Viennot et al. 2009, $http://www.geosciences.ens.fr/aqui-fr/\#_Documents$), compared to the 3 layers used in Modcou (Rousset et al., 2004). AquiFR has therefore a more realistic representation of groundwater and water exchanges between groundwater and rivers compared to the version of Modcou used in Bonnet et al. (2017). Water abstractions can be taken into account in AquiFR. However, in order to focus only on climate influences, the reconstruction developed in this study does not take water abstractions into account, which may explain some of the differences between the reconstruction and observations. In any case, water abstractions of the late 19th centuries are not known.

The Safran analysis, based on thousands of observation stations collected by Météo-France and an optimal interpolation algorithm, provides the seven atmospheric variables necessary to force the Surfex-AquiFR system (liquid and solid precipitation, incoming longwave and shortwave radiation fluxes, 10m wind speed, 2m specific humidity and temperature), at the hourly time step, on an 8km grid, from 1958 to present. The Safran analysis is described and evaluated in Quintana-Segui et al. (2008) and Vidal et al. (2010).

A simulation based on the Safran-Surfex-AquiFR system is available over the period 1958-present. This so-called reference simulation is used for the evaluation of the hydrometeorological reconstruction on their common period. As they share the same hydrological model, potential differences between the reconstruction and the reference simulation only depend on the quality of the reconstructed meteorological forcing.

## 2.3 Method

In order to extract the multidecadal variations from interannual series, a Lanczos low-pass filter (Duchon, 1979) with a cutoff frequency of 1/30 years and 31 weights is applied. No padding at the ends of the series is applied: the first 15 years and the last 15 years of the unfiltered series are considered as missing values in the filtered series. As this study is focused on multidecadal variations, long-term trends are first removed for most of the analyses presented in this study. As it is very unlikely for the trend to be linear on the long periods analyzed in the paper, non-linear trends are computed using an ensemble empirical mode decomposition algorithm (EEMD; Wu and Huang 2009). This algorithm decomposes a series into a sum of signals that characterize the different temporal variability scales that compose this series. The algorithm also extracts a non-linear trend. This method is adaptive to the signal studied and non-parametric.

In order to avoid an artificial skill induced by the annual cycle when calculating daily or monthly correlations between the observations and the reconstruction, the series are deseasonalized. For daily series, a centered running average of 31 days is applied first to limit the noise in the computed annual cycle. Then, the climatological average is calculated for each day of the year. For monthly series, the annual cycle is simply calculated as the climatological average of each month.

In this paper, winter means December-January-February (DJF), spring means March-April-May (MAM), summer means June-July-August (JJA) and fall means September-October-November (SON).

## 3  Development of the Seine hydrometeorological reconstruction

A new hydrological reconstruction, based on hydrological modelling, is developed over the Seine basin, improving the method presented in Bonnet et al. (2017), with two main objectives: (i) to extend the study period to the 1850s, in order to characterize more robustly multidecadal hydroclimate variations, and (ii) to improve the representation of river flows, particularly at the daily time scale, in order to obtain a better representation of high and low flows and study their multidecadal variations. Figure 2 describes the main steps of the method developed in the present study and highlights the improvements over the one used in Bonnet et al. (2017).

To obtain the meteorological forcing necessary for hydrological modelling, the main idea of the Bonnet et al. (2017) method is to use the analog method (Lorenz, 1969), a stochastic statistical downscaling method, to downscale a long-term atmospheric reanalysis (e.g. NOAA 20CR) and produce an ensemble of trajectories of precipitation and temperature over France (Step 1, Figure 2). Then, local long-term monthly precipitation and temperature observations are used to select the best trajectory (Step 3, Figure 2). In the present study, we add a daily constraint to this method in order to improve the simulation of daily river flows (Step 2, Figure 2).

To extend the study period, the long-term NOAA 20CRv2c atmospheric reanalysis (Compo et al., 2011), which begins in 1851, is used. This reanalysis is based on a global atmospheric model, using observed sea ice and sea surface temperature (SST) as boundary conditions together with the assimilation of surface and sea level pressure observations. The SSTs used are SODAsi.2 (Giese et al., 2016) for latitudes between 60N and 60S and COBE-SST2 (Hirahara et al., 2017) for the highest latitudes. 56 members, sampling the reanalysis uncertainties, are available.

## 3.1 step 1: Statistical downscaling

The analog method is based on the hypothesis that two days with similar large scale atmospheric states (e.g. large scale atmospheric circulation over the North Atlantic) are characterized by similar local weather conditions. In its most basic form, for each day $D$ of the reanalysis, the day $D_a$ (the so-called analog day), with the closest large scale atmospheric state is searched within the learning period, defined as the common period between the reanalysis and the observational database with the local variables necessary for hydrological modelling, e.g. here the Safran analysis. The local variables of interest of the day $D_a$ in the observational database are selected as an estimate of the local weather conditions for the day $D$. To quantify the similarity between large scale atmospheric states, four predictors are used in the present work: precipitation, surface temperature, sea level pressure and specific humidity at 850 hPa. The Euclidean distance is computed for each predictor, except for sea level pressure, for which the Teweles and Wobus score (Teweles Jr and Wobus, 1954; Obled et al., 2002) is calculated. The distances and the score are then combined after standardization to give the same weight to each predictor. Two domains of analogy are used. The domain for sea level pressure is delimited by the following coordinates: 44°N, 56°N, -11°E, 16°E. The domain for the three other predictors is defined by 46°N, 51°N, -2°E, 7°E.

Compared to Bonnet et al. (2017) where only one member of the long term reanalysis is downscaled (NR=1, Figure 2), here we statistically downscale the 56 members of NOAA 20CRv2c (NR=56, Figure 2). It leads for each day $D$ of the reconstruction period to a much larger pool of potential analog days, which allows to add a new step: a daily constraint by local observations (Step 2, Figure 2). The objective of this new step is to obtain a better representation of daily variations of the meteorological forcing.

## 3.2 step 2: Daily constraint

Instead of searching only for the best analog day $D_a$ for each day $D$ of the reconstruction period (1852-2008), the N best analog days $D_{a1}$, $D_{a2}$,...$D_{aN}$ in the learning period (1958-2008, limited by the availability of Safran) i.e. with the most similar large-scale atmospheric states are searched. In the present method, $N = 50$. As the 56 members of NOAA 20CRv2c are downscaled, in the end 2800 potential analog days are obtained for each day $D$ of the reconstruction period (with potentially similar analog days for the different members). As each analog day corresponds to a day of the learning period, the corresponding daily maps of precipitation $Pr(D_{a1})$, $Pr(D_{a2})$,...$Pr(D_{aN})$ and temperature $Tas(D_{a1})$, $Tas(D_{a2})$,...$Tas(D_{aN})$ from Safran are selected and compared to the daily station observations (SQR, see section 2.1) after regridding. Regridding consists in selecting the Safran grid point closest to each observation station over the Seine basin. Note that the number of stations varies over the 1852-2008 period. The comparison is therefore done each day of the reconstruction based on the available stations.

The daily comparison is based on the following approach:

(i) The average daily bias in mean precipitation averaged over the Seine basin is calculated for the 2800 analog days, and the 60 analog days with the lowest bias are selected.

(ii) The spatial root mean square errors for the 60 analog days are calculated for temperature. For precipitation, the error to the cubic power rather than to the square power is used, in order to give more weight to strong values of precipitation, and the absolute value is used.

(iii) The daily series of spatial errors obtained for precipitation and temperature are then standardized based on the statistics of the entire period and added, with a weight of 1 for precipitation and 0.5 for temperature.

(iv) Finally, each day of the reconstruction period, the 3 best analog days (out of 60), i.e. with the smallest errors compared to observations, are selected.

## 3.3 Step 3: Monthly constraint

Based on the 3 analog days selected thanks to the daily constraint, multiple trajectories of precipitation and temperature over the Seine basin are then created by repeatedly selecting randomly one of the 3 analog days for each day $D$ of the reconstruction. In practice, 5000 trajectories are created (Step 3, Figure 2). The monthly averages of precipitation (temperature) for these trajectories are computed. From the 3 different maps of precipitation (temperature) over the Seine basin obtained with the daily constraint for each day $D$ of the reanalysis, 5000 different monthly maps of precipitation (temperature) are obtained with this procedure.

For each month of the reconstruction, the 5000 maps of precipitation (temperature) obtained on the Safran grid are re-gridded and compared to the actual observed precipitation (temperature) map, using the long-term homogenized precipitation (temperature) series over France (see section 2) as reference. The spatial root mean square errors (RMSE) are computed for temperature and precipitation. The sum of the RMSEs corresponding to precipitation and to temperature is then computed, after the temporal standardization of the series of RMSEs in order to give the same weight to each variable. In the end, for each month of the reconstruction, the daily series of analog days among the sample of 5000 that leads to the lowest sum of RMSEs is selected. The term "monthly constraint" used in this study refers to this last step (Step 3 in Figure 2).

The interest of using a monthly constraint after the daily constraint is that monthly data are homogenized contrary to daily data (Section 2) and therefore it allows for a better representation of low-frequency variations.

## 3.4 Discussion

The method described above benefits from the advantages of the analog statistical downscaling method. Thanks to the analog days, all the meteorological variables from Safran necessary to force the Surfex-AquiFR hydrological model are obtained. The spatial and inter-variable consistencies are maintained after this procedure, because for each day of the reconstruction the entire map of precipitation (and temperature, humidity etc.) over France from Safran is selected based on a single analog day. Compared to a basic statistical downscaling method, our method additionally takes into account local observations in the downscaling process and not simply large scale information. The use of local observations allows for a more accurate reconstruction, as shown in Bonnet et al. (2017). Note that the temporal consistency of the meteorological forcing is ensured by both the temporal consistency of the predictors and of the local observations.

Multiple tests have been conducted to set-up the different ad-hoc aspects of the method, trying to obtain the best overall hydrometeorological reconstruction. These tests concern, for example, the best combination of weights given to precipitation and temperature errors at step 2 or the number of analogs selected at each step. For example, selecting only the 3 best analog days at the end of step 2 leads to best overall performance in capturing daily and monthly variations. Using more analog days may allow for a better representation of monthly variations but degrades the representation of daily variations.

To sum up, the hydrometeorological reconstruction developed over the Seine basin is constrained on a daily basis over the period 1885-2003 by observations of precipitation and temperature (SQR), on a monthly basis over the period 1885-2005 by homogenized observations of precipitation and temperature (SMR), and over the 1852-1884 and 2005-2008 periods by the monthly series of precipitation at Paris (Slonosky, 2002) (see section 2.1 for more details). The results, especially at the daily time scale, have therefore to be interpreted with more caution over the period constrained only by the monthly series of precipitation.

During the development of the reconstruction, mean climatological biases were found for reconstructed precipitation and incoming shortwave radiation in comparison to Safran over their common period. These mean climatological biases are corrected before hydrological modeling by calculating for each season of the study period the climatogical bias using Safran as reference. Then, for each season (e.g. winter 1852, spring 1852, summer 1852...), the bias is corrected proportionally at the hourly time step.

The meteorological forcing obtained for the 1852-2008 period with the approach described in this section is finally used to force the Surfex-AquiFR hydrogeological model to obtain the hydrological reconstruction over the Seine basin (Step 4, Figure 2).

## 4    Evaluation of the Seine reconstruction

### 4.1    Mean river flows

First, the temporal correlations between observed river flows and river flows from the Bonnet et al. (2017) reconstruction , the present reconstruction and the reference simulation are calculated at the daily and monthly time scale for the 1958-2005 period at the 136 gauging stations available over the Seine basin (Section 2.1). The median of these temporal correlations is respectively of 0.7 and 0.97 for daily and monthly time scales (Figure 3). The daily and monthly river flow variations are therefore correctly captured in the Seine reconstruction. At the monthly time scale, the river flow variability is reproduced almost as effectively as in the reference simulation. For most gauging stations, the representation of daily and monthly river flows is improved in the Seine reconstruction compared to the reconstruction developed over France by Bonnet et al. (2017). Some low correlations are still seen for a few stations (Figure 3). These stations are identical in the reconstructions and in the reference simulation, showing that this issue is not due to the reconstructed meteorological forcing. Non-climatic anthropogenic influences (e.g. dams, pumping or land use changes) that are not taken into account by the hydrological model, or measurement artifacts might be responsible for these lower correlations. Alternatively, they may be due to the hydrological model, especially

since these lower correlations are seen for small catchments (not shown), generally less well simulated by the hydrological system.

To evaluate the reconstruction on a longer period, river flows from the reconstruction are compared to observations at the two stations with long-term data on the catchment: the Seine at Paris and the Armançon at Aisy-sur-Armançon (Section 2) (Figure 4a-b). The reconstruction correctly reproduces the interannual variability at both stations, although differences in terms of amplitude are seen. For example, river flows in 1910, influenced by an exceptional flood, are underestimated at Paris. They are more correctly reproduced at Aisy-sur-Armançon. The multidecadal variability of observed flows is also quite well reproduced by the reconstruction (Figure 4c-d), even if the anomalies for the Seine at Paris tend to be underestimated at the beginning of the 20th century and overestimated at the end (Figure 4c). Differences in long-term trends, with a stronger negative trend in the observed series compared to the reconstruction (not shown), are probably at the origin of these differences. It is very unlikely that a potentially unrealistic trend has been introduced in hydrological reconstruction by the meteorological forcing developed in section 3, as the reconstructed precipitation and temperature series do not show unrealistic trends (not shown). It is possible that measurement artifacts, or non-climatic anthropogenic influences, not taken into account in the reconstruction, have influenced the long-term variations of observed river flows in that region heavily impacted by human activities. Multidecadal river flow variations at Paris and at Aisy-sur-Armançon are generally in phase. A strong positive multidecadal phase, which corresponds to a succession of years with higher than average river flows, is visible around 1920. On the contrary, a negative phase is present around 1890 and 1960.

## 4.2   Annual maximum daily river flow

The annual maximum of daily river flows are computed for the Seine reconstruction, the reference simulation and the observations for the Seine at Paris and compared (Figure 5). The interannual variability of the annual maximum daily river flow of the Seine at Paris is well captured in the Seine reconstruction, with a correlation of 0.88 with the observations on the 1900-2005 period (Figure 5). The correlation is consistent over time, with a correlation of 0.92 over the 1900-1960 period, and of 0.89 over the 1960-2005 period. For the later period, the correlation between the reconstruction and the observation is close to the correlation obtained with the reference simulation (r=0.95).

The annual maximum daily river flow is generally underestimated in the reconstruction before the 1960s (Figure 5). As a result, the magnitude of the exceptional 1910 flood is largely underestimated in the reconstruction, even if this event remains the largest in the reconstruction. The bias is far less pronounced in the reconstruction after 1960 and in the reference simulation, as a result of an apparent decrease in the observed annual maximum. The construction of large reservoirs over the Seine basin upstream from Paris that started in the 1960s, whose objectives include flood control, might be the cause of the observed decrease.

## 4.3   Piezometric levels

In France, long-term observations of piezometric levels (measure of the head of a groundwater table) are very rare. The measurements at the Toury sugar plant, which have monitored the Beauce groundwater table from the beginning of the 20th

century, is especially interesting in this context. The Beauce aquifer is a limestone aquifer from the Oligocene, and it is drained by rivers mainly at its borders. The observed piezometric level at Toury shows strong multidecadal variations, with a positive phase around the 1920-1940 period, preceded by a negative phase in the early 20th century and followed by a negative phase in the 1950s (Figure 6). Strong decadal peaks are also seen after 1950, consistent with Flipo et al. (2012). The evolution of the Beauce groundwater is generally consistent with that of river flows (Figure 4) at multidecadal time scales.

The interannual variations of piezometric heads at Toury are quite well reproduced in the reconstruction, with a correlation of 0.74 over the 20th century. The amplitude of these variations is however strongly underestimated, by a factor of 2, both in the reconstruction and in the reference simulation. This might be linked to a coarse representation of the aquifer complexity in AquiFR. A partially captive part of the limestone aquifer, which contains lenses of clay, is not represented in the model due to his insufficient resolution. This part of the aquifer amplifies the flow time and, therefore, the memory of the aquifer.

## 5 Multidecadal hydroclimate variations in the Seine basin

### 5.1 Characterization

Low-pass filtered annual river flows of the Seine at Poses (see location in Figure 1) from the reconstruction show strong multidecadal variations during the 20th century (Figure 7), consistent with the observed variations over France described in Boé and Habets (2014).

Major sampling uncertainties exist when dealing with multidecadal variations using the short observational record. Additionally, since the mid-20th century, the climate has been strongly impacted by anthropogenic forcings, making it difficult to disentangle the respective role of internal variability and external forcings in observed hydroclimate variations. Thanks to the extended length of our reconstruction compared to previous works, we show that multidecadal variations also exist before the 20th century, with lower than average river flows around the 1885-1905 period (about 15% lower than the average of 446m3/s for annual river flows). This reinforces our confidence in the reality of the multidecadal hydrological variations in France observed over the 20th century and in the idea that they are at least partly of internal origin. Phased multidecadal variations also exist for the seasonal averages, with the strongest absolute variations seen in spring and winter. Note however that as climatological river flows are smaller in summer and early fall, the multidecadal variations seen in these seasons are also important as shown later in Figure 8.

To better understand the mechanisms at the origin of the multidecadal hydroclimate variations in the Seine basin, a composite analysis between the negative multidecadal phases, which correspond to the drier than average periods identified from the Seine river flows at Poses after low-frequency filtering (Figure 7), and positive multidecadal phases, which correspond to the wetter than average periods, is conducted for the main hydrological variables (Figure 8). The negative phases are defined as the 1885-1905 and 1940-1960 periods, and the positive phases are defined as the 1910-1930 and 1975-1995 periods. The use of four 20-year periods thanks to the length of the reconstruction makes the analysis of these variations more robust in comparison to previous works (e.g. Boé and Habets 2014 and Bonnet et al. 2017). A Student's t-test with a p-value < 0.05 is applied to evaluate the significance of changes between the negative and positive multidecadal phases.

Strong river flow variations between the negative and positive multidecadal phases are seen at most simulated stations (Figure 8a-e-i-m). The relative differences are stronger in spring and summer, as large as 40% for some stations, and significant almost everywhere.

In spring, the river flows anomalies are concomitant with strong and significant precipitation anomalies over the entire Seine basin (Figure 8f). During the negative multidecadal phases, spring precipitation is on average 20% lower than during the positive phases. These precipitation anomalies also lead to significant soil moisture anomalies, between 5 and 10% (Figure 8g). The evapotranspiration in spring does not show significant differences between the negative and positive multidecadal phases of the Seine river flows (Figure 8h), likely because evapotranspiration is energy-limited rather than water-limited in this region in spring.

In summer, unlike in spring, no significant differences in precipitation between positive and negative multidecadal phases are noted (Figure 8j). The significant summer river flow variations therefore cannot be explained by concomitant precipitation variations alone, suggesting that hydrological processes are at play. Large soil moisture anomalies, significant over a large part of the catchment, are noted in summer between positive and negative multidecadal phases (between 10 and 15%, Figure 8k). The absence of significant precipitation and evapotranspiration anomalies in summer suggests that the persistence of soil moisture anomalies from spring to summer (e.g. Boé and Habets 2014) is responsible for the multidecadal variations of summer soil moisture.

In winter, the relative variations in river flows between the positive and negative multidecadal phases are weaker than in spring and summer, but they are still significant for most of the stations. Slight differences in precipitation, between 5 and 10% on average over the Seine basin, are noted, but they are not significant. The variations in precipitation alone therefore likely cannot explain such river flow variations in winter. The negative multidecadal phases of the Seine river flows are characterized by significantly lower evapotranspiration anomalies (Figure 8d). Negative temperature anomalies, close to -0.8K and significant over the whole basin (not shown), are likely the cause of these evapotranspiration anomalies. As the evapotranspiration anomalies are negative during the dry phases, they are not responsible for the river flows anomalies and they actually tend to mitigate them. The soil moisture presents significant variations between the negative and positive phases over a part of the Seine basin in winter (between 5 and 10%) (Figure 8c). They are probably related to the differences in precipitation previously noted, even if they are not significant. In fall, only a few stations show significant differences in river flows between the positive and negative multidecadal phases, although some stations, mainly in the south east of the catchment, still show strong variations of 30% (Figure 8m). A part of these anomalies are likely driven by the concomitant variations in precipitation, significant over a small area in the south east of the catchment (Figure 8n).

Two important conclusions arise from the previous analysis. (i) Significant multidecadal variations in soil moisture exist in spring and summer from the mid-19th century, with potential important societal consequences, for example regarding agriculture, which is highly developed over the Seine catchment. (ii) Large and significant multidecadal variations in summer and winter river flows exist, which cannot be explained by concomitant variations in precipitation. Hydrological mechanisms that could be responsible for these variations are now investigated.

## 5.2    Role of groundwater table

In the Seine basin, the groundwater-river exchanges play an important role in sustaining summer flows (Rousset et al., 2004). The groundwater-river exchanges from the reconstruction, calculated by the AquiFR hydrogeological model, show strong multidecadal variations over the past 150 years for all seasons (Figure 9). These variations are in phase with the multidecadal variations in river flows noted previously and their amplitude is large (compare Figure 7 and Figure 9), pointing to an important role of groundwater dynamics in modulating multidecadal river flows. We calculated the standard deviation of filtered river flows series (as shown in Figure 7) before and after the subtraction of groundwater-river exchanges shown in Figure 9. The standard deviation is 64% lower in summer, 52% lower in autumn, 41% lower in spring and 38% lower in winter after the subtraction of the contribution of groundwater to river flows. Groundwater-river exchanges are therefore the dominant driver of multidecadal variations in summer and autumn river flows, with also an important influence in winter and spring. The multidecadal variations in groundwater levels are very likely due to variations in the recharge explained by the multidecadal variations noted previously in spring precipitation, especially in early spring, a period favorable to the recharge.

## 5.3    Role of soil moisture

As variations in groundwater-river exchanges, very likely induced by spring precipitation, roughly explain two third of the strong multidecadal river flow variations that exist in summer, other mechanisms are involved. As seen in Figure 8k, a large part of the basin in summer is characterized by significant negative soil moisture anomalies during negative multidecadal river flows phases, according to a Student's t-test with a p-value < 0.05. Soil moisture anomalies may impact river flows through the modulation of the partitioning of precipitation between runoff and infiltration (e.g. Boé and Habets 2014), as higher soil moisture may favor runoff over infiltration. To assess whether soil moisture may play a role in summer river flow variations, the relative changes in the ratio of total runoff to precipitation between the positive and negative multidecadal phases is calculated in summer with the reconstruction (Figure 10). During the dry multidecadal phases, the ratio between total runoff and precipitation in summer is significantly lower over a large part of the basin, where the soil moisture anomalies are significant (Figure 10). As suggested by Bonnet et al. (2017) for the Loire Basin (see Figure 1), positive multidecadal anomalies in spring soil moisture due to anomalous precipitation persist in summer, especially over the upstream part of the catchment, and then influence summer river flows by favoring runoff against infiltration.

## 6    Role of large-scale circulation and influence of ocean variability

Strong multidecadal hydroclimate variations in the Seine basin from the 1850s have been highlighted in the previous section. They are mainly linked to variations in precipitation in spring, and, to a lesser extent, in fall and winter. The question is now, therefore, to understand the origin of these precipitation variations. The role of large scale atmospheric circulation, an important driver of precipitation is first studied. A composite analysis of sea level pressure during the positive and negative multidecadal

phases of the Seine river flows previously defined is conducted. As in section 5.1, the significance of changes between negative and positive multi-decadal phases is assessed with a Student's t-test and a p-value < 0.05.

Sea level pressure shows significant differences between the positive and negative hydroclimate phases identified in section 5.1 (Figure 11a). The negative multidecadal phases of the Seine river flows are characterized by significantly higher pressures over northern Europe, and significantly lower pressures over North Africa. This circulation pattern leads to weaker westerlies over Western Europe and therefore to negative precipitation anomalies over the Seine basin. This dipole pattern of sea level pressure bears resemblance with the North Atlantic Oscillation (NAO), but with a slight shift to the south.

Due to its the chaotic nature, the existence of significant atmospheric circulation anomalies for a few decades suggests the existence of a slow external forcing acting on the atmosphere. Because of its inertia, the ocean could be involved. Significant variations of sea surface temperature in the North Atlantic are visible between the multidecadal phases of the Seine river flows. During the negative phases, the North Atlantic basin is characterized by warmer surface temperatures relative to the positive phases, around 0.4 to 0.7°C, especially in the subpolar gyre and the tropical North Atlantic (Figure 11b). These sea surface temperature (SST) anomalies in the North-Atlantic are reminiscent of those associated with the Atlantic Multidecadal Variability (AMV), the main mode of multidecadal variability in the North-Atlantic / Europe region. Our results are therefore consistent with Sutton and Dong (2012) who suggested, based on an observational analysis, that the AMV is associated with negative precipitation anomalies over France in spring, the season during which the decadal precipitation anomalies are the largest (Figure 8f), through a modulation of large scale atmospheric circulation. Boé and Habets (2014) came to a similar conclusion, but suggested the existence of a lag between the AMV and river flows anomalies, with the AMV leading by several years. Interestingly, considering a 10-year time lag in the composite analysis (with the SST leading), SST anomalies in the North Atlantic are much higher than with no lag (Figure 11c).

Significant SST anomalies between the negative and positive multidecadal phases of the Seine river flows are also observed in the North Pacific. SSTs there are significantly warmer during the negative multidecadal phases of Seine river flows compared to the positive phases. The potential mechanisms linking the SST anomalies in the North Pacific and multidecadal hydroclimate variations over France are not clear. Ding et al. (2017) suggest that the Pacific Decadal Variability (PDV) (Mantua and Hare, 2002) combined with La Niña events could result in precipitation and temperature anomalies over Europe by favoring NAO-like circulation patterns. The phase of the NAO and therefore the sign of precipitation and temperature anomalies is dependent on the type of La Niña. Note also that the apparent link between the North Pacific SSTs and Seine river flows might not be causal. SST anomalies in the North Pacific might be to a certain extent also driven by the AMV. Ruprich-Robert et al. (2017) indeed suggest that the AMV may influence North Pacific SSTs, mainly through an atmospheric connection.

In order to further investigate the relationship between the AMV and the multidecadal hydroclimate variations over the Seine basin, the lagged correlations between the AMV index and spring river flows and precipitation series are computed (Figure 12). The AMV index is defined as the low-pass filtered average of the North Atlantic SSTs, from which the forced signal has been removed (Deser et al., 2010). Here, the forced signal is estimated with the trend calculated with the EEMD method (see section 2.4). A "phase-scrambled bootstrapping test" with a p-value < 0.05 is used to assess the significance of the correlation Davison and Hinkley (1997). Significant anti-correlations (around -0.8 and -0.9) are found between the AMV index calculated

with the SSTs from the 20CRv2c reanalysis and low-frequency filtered spring precipitation over the Seine catchment, with a lag of about 10 years, consistent with the lag found by Boé and Habets (2014) (Figure 12a). Similar results are obtained using the AMV index from Wang et al. (2017). This paleoclimate reconstruction, based on multiple proxies, is largely independent of the first AMV index. Given the length of the paleoclimate AMV index of Wang et al. (2017) and the availability of a very long observed series of precipitation at Paris (Section 2), it is possible to investigate the temporal stability of the AMV / spring precipitation relationship. Similar results are obtained for the 1779-1889 and 1890-1990 periods (Figure 12a), highlighting the robustness of the relationship.

The influence of the AMV on spring precipitation is reflected for spring river flows. Significant anti-correlations are found between the AMV and the river flows of the Seine at Paris at multidecadal time scales for the 1876-1985 period, with a lag of about 10 years, as for precipitation (Figure 12b). The results obtained with the observations and the reconstruction are very similar.

The 20th century is a very short period to characterize multidecadal climate variations. On such a short period, the sampling uncertainties are very large, which may question the robustness of the results (Qasmi et al., 2017). Additionally, non-stationnarities may exist (Cassou et al., 2018) and the strong influence of anthropogenic forcings on the climate of the second half of the 20th century may complicate the interpretation of observed multidecadal variations. The results previously described are therefore important because they confirm those obtained in previous works, but on a considerably longer period, at least from the end of the 18th century. They show the robustness of the teleconnection between the AMV and spring precipitation over France described in Sutton and Dong (2012) and confirm the existence of the lag suggested in Boé and Habets (2014), even if its physical origin is still unknown. These results also confirm that the multidecadal hydroclimate variations observed over France are likely not mainly the result of anthropogenic forcings.

## 7  Influence of multidecadal river flow variations on high and low flows

### 7.1  Links with high and low flows

The improvements to the reconstruction described in section 3 allows for a better representation of high-frequency variations compared to previous works (Bonnet et al. 2017, see Figure 3) and therefore of low and high flows. To assess the influence of multidecadal hydroclimate variations on high and low flows, the ratios of the number of high (low) flow days during the positive and negative multidecadal phases of the Seine river flows are calculated (Figure 13). A high flow day is simply defined here as a day with flows larger than the 95th percentile computed on the four multidecadal phases considered (1910-1930, 1975-1995, 1885-1905 and 1940-1960) for the flood season, from November to April. Conversely, low flow days are defined as days with flows lower than the 5th percentile on the same periods but considering all months of the year.

Days with high flows are on average twice as likely to occur during a positive multidecadal phase of the Seine river flows, and up to three times more likely for some stations (Figure 13a). During the flood season, the positive multidecadal phases of the Seine river flows are characterized by higher mean precipitation, wetter soils (Figure 8) and higher groundwater levels (not shown). Such anomalies favor the occurrence of high flows. The positive multidecadal phases are also characterized by

more days with intense precipitation (not shown), which also favors the occurence of high flows. This result is consistent with Willems (2013), who highlighted the existence of multidecadal variations in observed rainfall extremes in Europe, correlated with intense river flows. The results from the reconstruction are consistent with the observations for the two long series available in the Seine basin (Seine at Paris and Armançon at Aisy-sur-Armançon, see triangles in Figure 13a).

The multidecadal hydroclimate variability of the Seine basin does not seem to strongly influence daily low flows for many stations. The ratio of low flow days between negative and positive multidecadal phases is indeed between one and two for a large number of stations (Figure 13b). For the observations at Aisy-sur-Armançon (triangle in the south east of the basin) the ratio is 1.1, consistent with the reconstruction. For the observations at Paris, days with low flows are more than four times more frequent during negative multidecadal phases, whereas they are only twice more frequent in the reconstruction. This discrepancy might be explained by the important influence of human activities on the Seine at Paris. Some stations, notably in the central part of the catchment show a ratio close or greater than two, pointing to an influence of multidecadal hydroclimate variations on low flow days there. It may be due to the multidecadal variations seen previously in groundwater-river exchanges (Figure 9), which may impact the daily low flows.

## 7.2 The 1921 and 1949 droughts

To further understand the influence of multidecadal hydroclimate variability on droughts, we focus on two major droughts of the last 150 years: the 1921 and 1949 events. The 1921 drought, characterized as the most severe hydrological drought of the Seine basin since the 1850s in our reconstruction (not shown), occurred in the middle of a positive multidecadal phase of the Seine river flows (Figure 7). The 1949 drought, characterized as the second longest and third strongest hydrological drought in the Seine basin according to the hydrometeorological reconstruction (not shown), occurred on the contrary in the middle of a negative multidecadal phase. Droughts are defined as the periods during which the VCN3 index (3-day moving average minimum flow) remains inferior to the 5-years return value. The severity of a drought is defined as the relative difference between the VCN3 index averaged over a given drought with the average value of the index for all droughts.

Considering the 20-year periods centred on these events, the mean climate conditions during which these events developed are therefore very different. Climatological groundwater levels are very high in the decades around 1921 compared to the decades around 1949 (Figure 14d), with therefore larger groundwater-river exchanges (Figure 14c). The same applies to river flows (Figure 14b).

For both droughts, precipitation is below the climatological average from the autumn of the previous year (i.e. n-1) and extremely small in February, March, April of the year n. Between October of year n-1 and December of year n, average precipitation is lower in 1921 than in 1949 (1.25 and 1.50 mm/day respectively, well below the climatological average in both cases). The groundwater levels in autumn 1920 are very high, partly because of the multidecadal variations previously described, while the heads are already below average in autumn 1948. In both cases, the levels fall during the drought, but they remain not extreme in 1921, while exceptionally low values are reached in spring 1949. As a result, the groundwater to river exchanges in 1921 remain above those of 1949 by between 17 and 83 $m^3$/s from August of year n-1 to September of year n (Figure 14c-d). Despite the lower groundwater-river exchanges, the rivers flows of the 1949 event generally remain above

those of the 1921 drought from the autumn of year n-1 to the spring of year n. The larger groundwater-river exchanges in 1921 contributed to mitigate the severity of the 1921 hydrological drought, while in 1949, the lower climatological groundwater levels contributed to increase the severity of the drought. The already exceptional 1921 drought could have been even worse had it occurred during a dry multidecadal phase as for the 1949 event.

## 8   Discussion

In this study, a new hydrometeorological reconstruction focusing on the Seine basin is presented. The method improves upon the one described in Bonnet et al. (2017). It leads to a better representation of daily river flows. This reconstruction is therefore suitable to study the multidecadal variability of high and low flows, as well as hydrological extremes. The reconstruction is extended back to middle of the 19th century. The longer study period compared to previous works (e.g. Boé and Habets 2014) allows for a more robust characterization of multidecadal hydrological variations, which is crucial in that context (e.g. Cassou et al. 2018).

Although the reconstruction developed in this study is an useful tool for studying the past variability of the hydrological cycle over the Seine basin, it is obviously not perfect. Uncertainties are present throughout the modelling chain. The statistical downscaling method used at the first step of the reconstruction method assumes that the learning period, over which the large-scale reanalysis and the Safran analysis overlap (1959-2010), is representative of the meteorological conditions of the 1851-2010 period. The consistent performances of the reconstruction over the entire period shown by several analyses in this study suggest that this hypothesis has no major impact on our results. Important uncertainties are associated with the 20CRv2c reanalysis at the beginning of the period, due to the smaller number of assimilated observations (Krueger et al., 2013). We use monthly homogenized local precipitation and temperature observations to constrain the results of statistical downscaling in order to improve the temporal homogeneity of the reconstruction, but the homogenization method is quite old and does not benefit from recent advances in homogenization procedures. The good agreement between the low-frequency variations of the homogenized monthly precipitation series and of the Global Precipitation Climatology Centre dataset (Schneider et al., 2008) from 1901 to 2011 (not shown) still gives good confidence in the overall realism of the multidecadal variations described in this study.

The hydrological model used in this work is also not perfect, with potential consequences on our results. The amplitude of the variations in the piezometric levels at Toury in particular is strongly underestimated. Although this is a one-point measurement, it is possible that this underestimation exists for the entire Beauce aquifer, which could imply an underestimation of the multidecadal variability of reconstructed river flows, especially in summer when the role of groundwater is particularly important. This underestimation is likely due to hydrological modelling, and could come from a too simple representation of the limestone aquifer that neglects some confined parts. The river channel dimensions and conveyance capacity are also supposed constant over time, whereas they could be affected by multidecadal climate variability (Slater et al., 2019).

Some important uncertainties are also associated with the observations used for the evaluation of the reconstruction. They may indeed be influenced by non-climatic anthropogenic influences such as dams and pumping, not taken into account in the

hydrological model, or changes in measurement methods for example. We have evaluated the reconstruction against multiple observations in an effort to ensure that our results are not too dependent on a particular source of information and to better understand the potential limitations of the reconstruction.

Even if it is not the object of this study, an important difference of long-term trends in river flows exists between the reconstruction and the observations for the Seine at Paris. The origin of these differences is not understood currently. The meteorological variables used as forcing for the reconstruction, and precipitation in particular, don't show unrealistic trends (not shown) and therefore the trends in reconstructed river flows may not be unrealistic. As the Seine basin is heavily impacted by human activities, it is possible that non-climatic anthropogenic influences, which are not taken into account in the reconstruction, are responsible for the large negative trend in the observed series. It is something worth investigating in future work.

## 9   Conclusions and perspectives

Based on the reconstruction developed in this study, we show the existence of strong multidecadal variations in river flows over the Seine catchment since the 1850s. Consistent with previous studies over France (Boé and Habets (2014) and Bonnet et al. (2017)), strong variations are noted in spring. For many stations, even stronger variations are noted in summer. Significant multidecadal variations are also noted in winter thanks to the longer period studied here. Spring precipitation directly impacts river flows in spring but also river flows in the other seasons through different hydrological processes. Spring precipitation leads to soil moisture anomalies in spring that persist until summer, regulating the runoff to precipitation ratio. Precipitation variations in early spring impact on groundwater recharge and then river flows during all seasons through groundwater-river exchanges. Both the multidecadal variations in soil moisture and in groundwater may have important direct societal impacts, independently of their impacts of river flows, in particular with regards to agriculture. Interestingly, the variations in summer river flows are much larger than seen in Bonnet et al. (2017). These differences are likely both due to the longer period studied here and to different hydrogeological models, with a finer representation of aquifers in the present study.

The multidecadal hydroclimate variations over the Seine basin are due to large-scale atmospheric circulation anomalies, themselves associated with sea surface temperature anomalies in the North Atlantic. Interestingly, a potential connection to the North Pacific, not described before, is also noted.

As discussed by Cassou et al. (2018), because of major sampling uncertainties and potential non-stationnarities, it is important to extend the analyses of multidecadal climate variations as far back in time as possible, in order to confirm or deny the results obtained by most studies on the 20th century. Our results confirm the link between the AMV and French hydro-climate variations seen in previous studies on shorter periods, pointing to its robustness. The fact that multidecadal hydroclimate variations exist before the 20th century, a period where the role of anthropogenic forcings is small, also reinforces the idea that they are of internal origin to a large extent and not primarily forced by greenhouse gases nor sulphate aerosols.

An interesting new result is that the multidecadal variations previously described for the mean hydroclimate may also influence high and low flows. The positive multidecadal phases seem to be more conducive of flooding. Over large parts of

the catchment the probability of high flows is more than doubled during wet multidecadal phases. Although these events are mainly the result of specific weather conditions, the mean state of the hydrological system (e.g. soil moisture or groundwater) influenced by the multidecadal hydroclimate variations, may amplify or reduce the magnitude of these events. The impact of multidecadal variations in low flows is more regional. Still, we have illustrated for two major droughts of the 20th century, in 1921 and 1949, how multidecadal hydroclimate variations, through their impact on groundwater, may modulate the intensity of droughts. Multidecadal variations in groundwater may have reduced the intensity of the 1921 drought and increased the intensity of the 1949 drought.

The results described in this paper illustrate that relying on short periods of few decades (e.g. three) to characterize hydrological hazards can be problematic, especially for practical purposes. Indeed, with such short periods, the existence of multidecadal hydrological variations may make the estimation of flood characteristics or of the probability of occurrence of droughts, for example, unreliable.

This study reinforces the idea that the hydroclimate variability over the Seine basin and more generally over France results from a teleconnection with North Atlantic SSTs, and suggests a potential role of North Pacific SSTs. The associated mechanisms are still poorly understood. The Decadal Climate Projection Project (Boer et al., 2016) of the Coupled Model Intercomparison Project Phase 6 (Eyring et al., 2016) proposes some dedicated numerical experiments to tackle these questions. They will hopefully allow some progress in our understanding of the physical origin of multidecadal hydrological variations over France.

Given the large multidecadal hydroclimate variations described in this study, it is crucial to assess whether climate models reproduce them correctly and therefore if the uncertainties due to internal variability are correctly captured in current projections. It may not be the case. For example, Qasmi et al. (2017) show that climate models may have difficulties capturing some properties of the AMV. Simpson et al. (2018) suggest that current climate models underestimate the multidecadal variations of the atmospheric jet over the North Atlantic in late winter / early spring, which would have consequences for the French hydroclimate. Understanding the potential consequences of these issues regarding the uncertainties in hydroclimate projections is now crucial.

*Competing interests.* The authors declare that there are no conflicts of interest regarding this work.

*Acknowledgements.* This work has been supported by the French National program LEFE/INSU through the VITESSE project. The figures have been produced using the NCAR Command Language: The NCAR Command Language (Version 6.3.0) [Software]. (2016). Boulder, Colorado: UCAR/NCAR/CISL/TDD. https://doi.org/10.5065/D6WD3XH5. The authors acknowledge all the colleagues from Météo-France and Mines-ParisTech who have been contributing to the development of the Safran-Surfex system, as well as Aqui-FR, and Météo France (direction de la climatologie et des services climatiques) for providing the precipitation and temperature data sets. We also want to thank Thierry Morel from Cerfacs, strongly involved in the Aqui-FR project. The authors also thank Laurence Lestel, involved in the VITESSE project, for useful discussions about this work. Twentieth Century Reanalysis data were provided by the NOAA/OAR/ESRL PSD, Boulder,

Colorado, USA, from their website: https://www.esrl.noaa.gov/psd/. Support for the Twentieth Century Reanalysis Project data set is provided by the U.S. Department of Energy, Office of Science Innovative and Novel Computational Impact on Theory and Experiment (DOE INCITE) program, and Office of Biological and Environmental Research (BER), and by the National Oceanic and Atmospheric Administration Climate Program Office. Other data used in this study are available from the authors upon request.

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

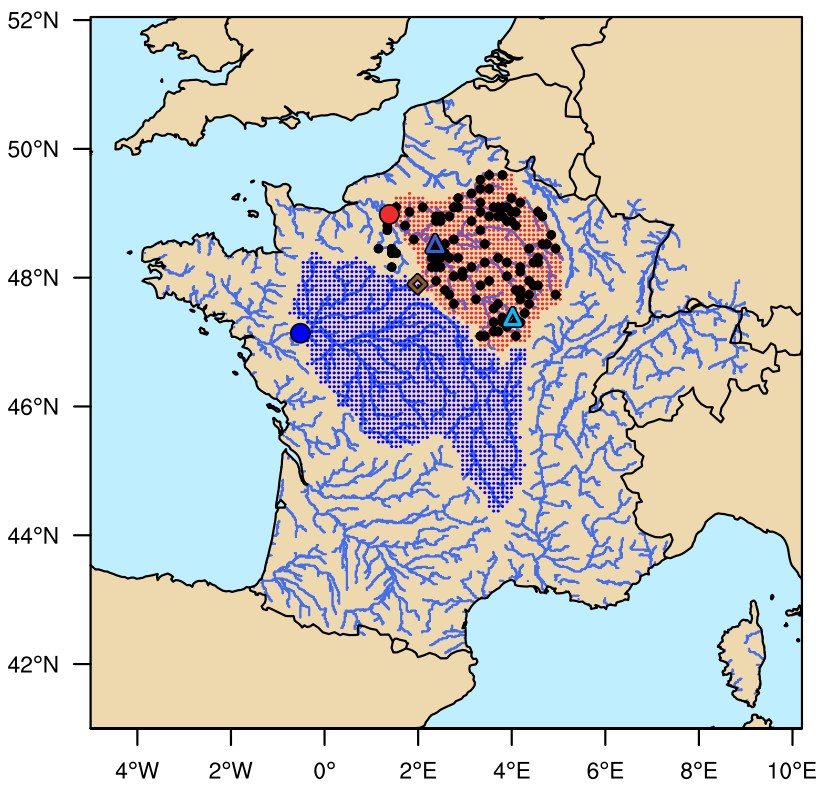

**Figure 1.** Location of the Seine catchment (red area) corresponding to the Seine at Poses station, considered as the outlet (red dot), and of the Loire catchment (blue area) for the Loire at Montjean sur Loire station (blue dot). Location of hydrometric stations both observed and simulated (black circles), and of the two long-term river flows observations available: the Paris Austerlitz station (dark blue triangle) and the Aisy-sur-Armançon station (light blue triangle). Location of the long-term piezometric level series at Toury (brown rhombus).

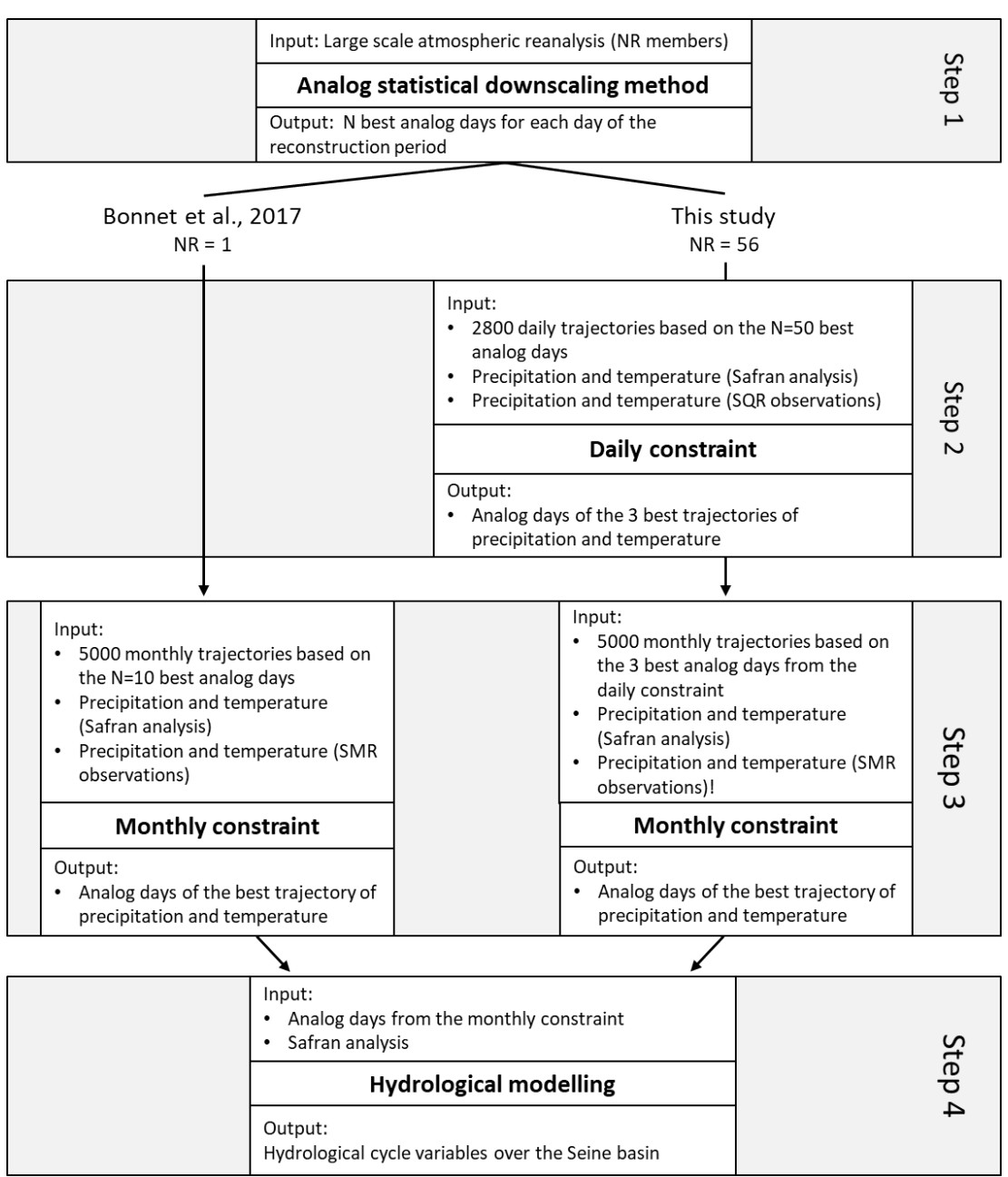

**Figure 2.** Schematic representation of (on the right) the method used to obtain the hydrological reconstruction in this study and (on the left) the method used in Bonnet et al. (2017), which doesn't include a daily constraint (Step 2).

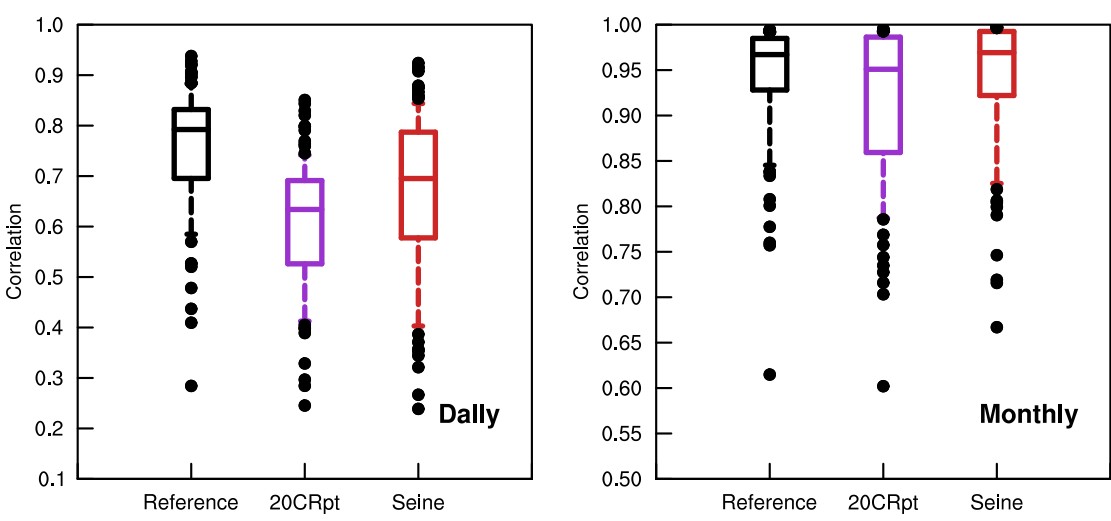

**Figure 3.** Spatial distribution of the correlations between the 136 river flow observations over the Seine basin at (left) daily and (right) monthly time steps and the (black) reference simulation (Safran-Surfex-AquiFR), (purple) the 20CRpt reconstruction described in Bonnet et al. (2017) and (red) the Seine reconstruction developed in the present study. The correlations are calculated on the 1958-2005 period and the series have been deseasonalyzed beforehand. The boxplots show the 10th percentile/25th percentile/median/75th percentile and the 90th percentile. The black dots are the values below/above the 10th and the 90th percentiles.

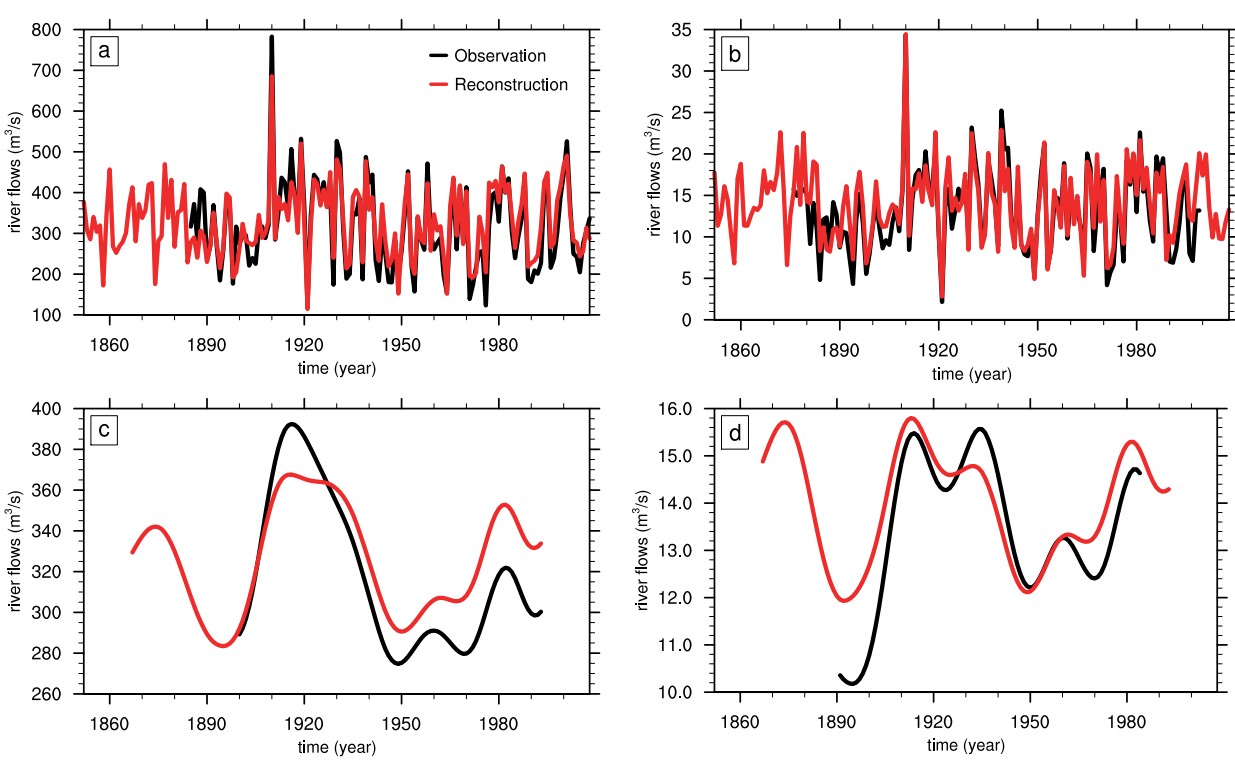

**Figure 4.** Annual series of river flows for (a) the Seine at Paris and (b) the Armançon at Aisy-sur-Armançon (see Figure 1). (c) and (d): annual low-pass filtered series of river flows for the same stations as (a) and (b). Black: observations. Red: Seine reconstruction.

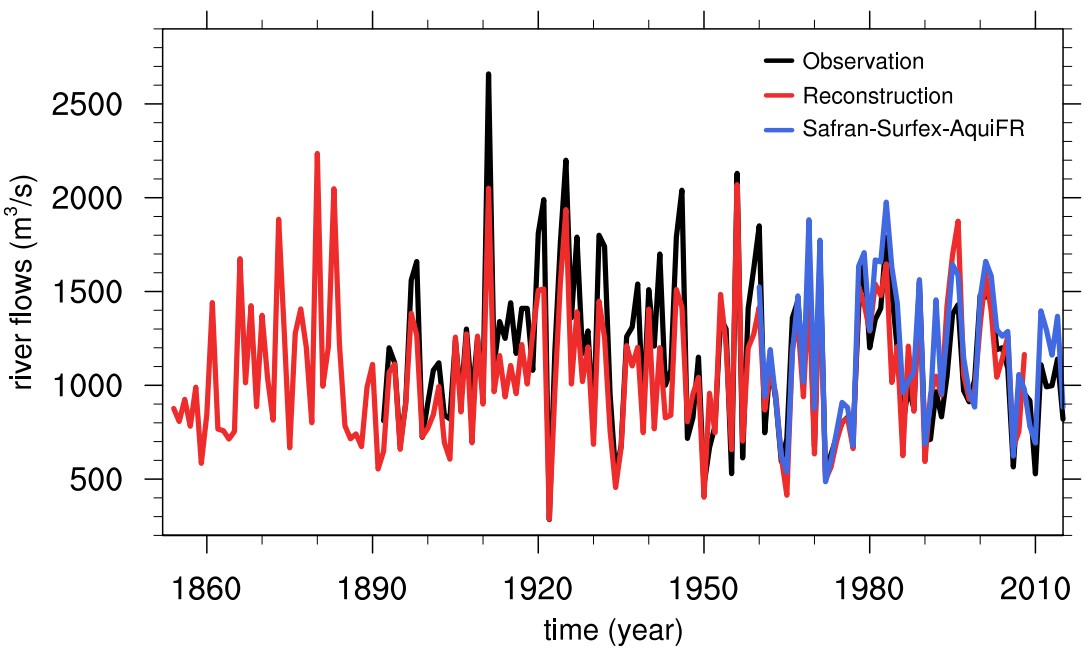

**Figure 5.** Annual maximum of daily river flows for the Seine at Paris. Black: observations. Red: Seine reconstruction. Blue: reference simulation (Safran-Surfex-AquiFR).

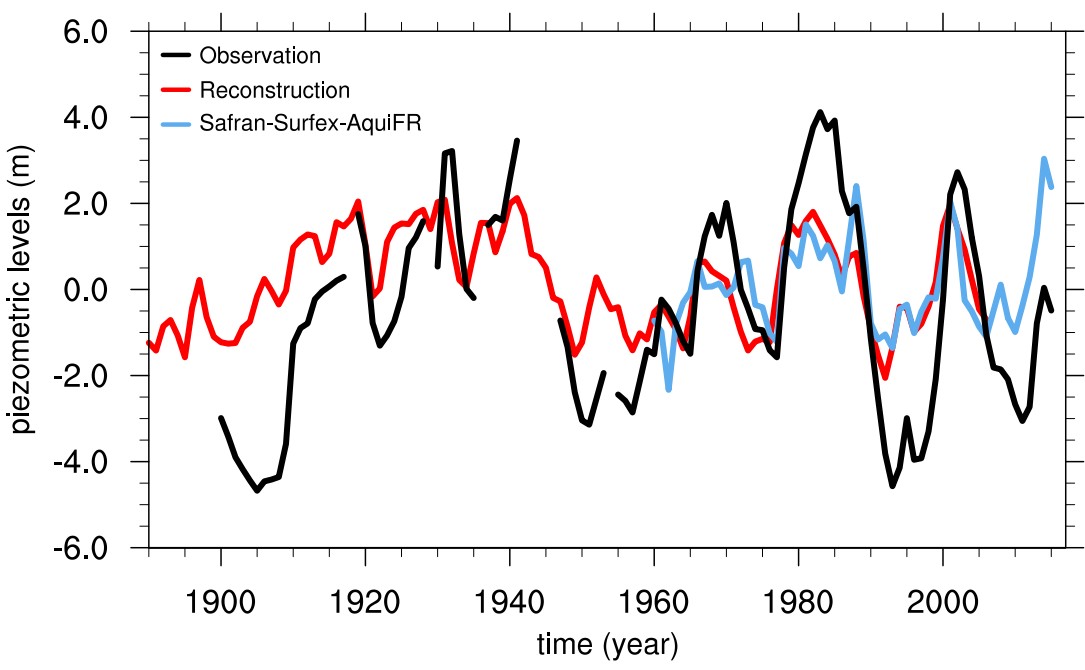

**Figure 6.** Annual anomalies of the piezometric levels of the Beauce groundwater table at Toury. The reference period is 1960-2005. Black: Observations. Red: Seine reconstruction. Blue: reference simulation (Safran-Surfex-AquiFR).

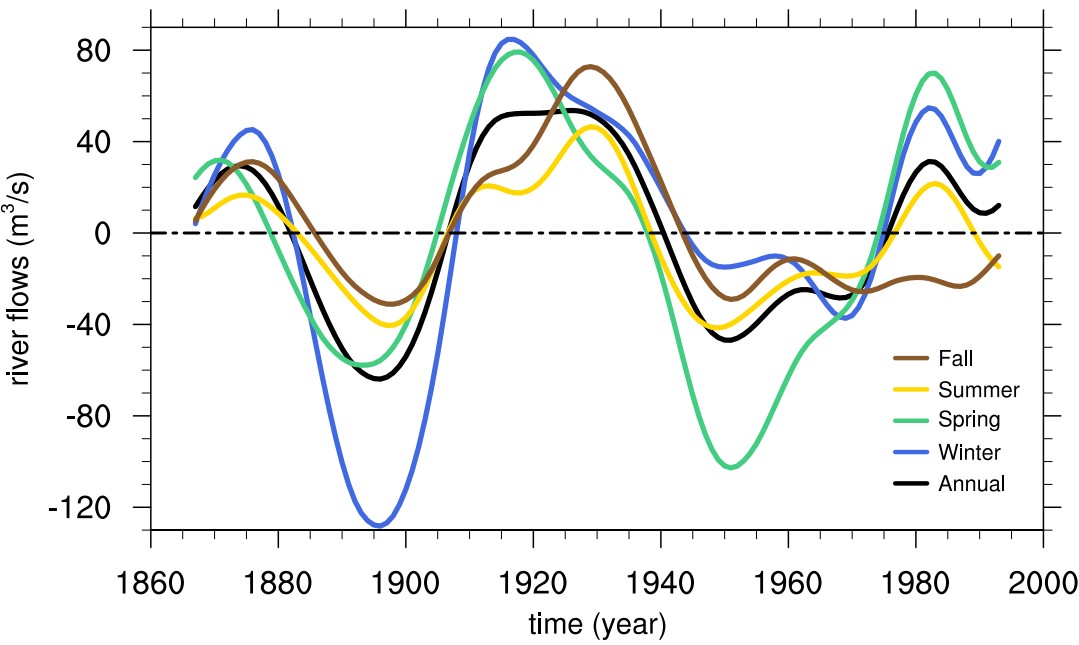

**Figure 7.** Low-pass filtered interannual anomalies of the Seine river flows at the Poses station in the reconstruction. Black: annual, blue: winter, green: spring, yellow: summer and brown: autumn. The annual mean over the whole period is 446m3/s.

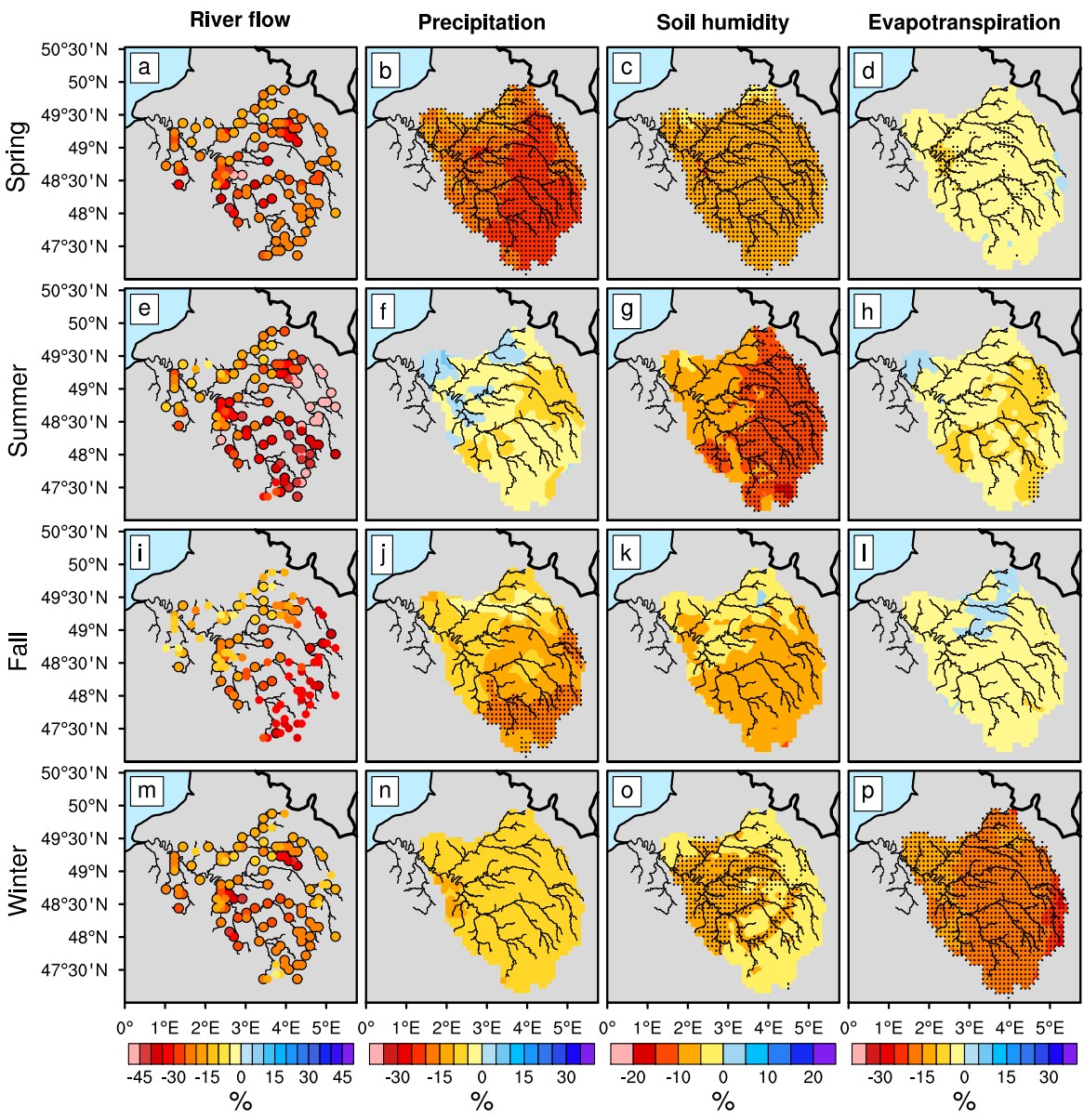

**Figure 8.** Relative differences (%) in detrended (a-e-i-m) simulated river flows, (b-f-j-n) precipitation, (c-g-k-o) soil wetness index (SWI) and (d-h-l-p) evapotranspiration, between the negative multidecadal phases of the Seine river flows (1885-1905 and 1940-1960) and the positive phases (1910-1930 and 1975-1995). The reference is calculated as the average of these four periods. (a-b-c-d) winter, (e-f-g-h) spring, (i-j-k-l summer) and (m-n-o-p) autumn. Simulated river flows, precipitation, SWI and evapotranspiration come from the Seine reconstruction. Black circles and dots show where the differences are significant based on a Student's t-test with a p-value < 0.05.

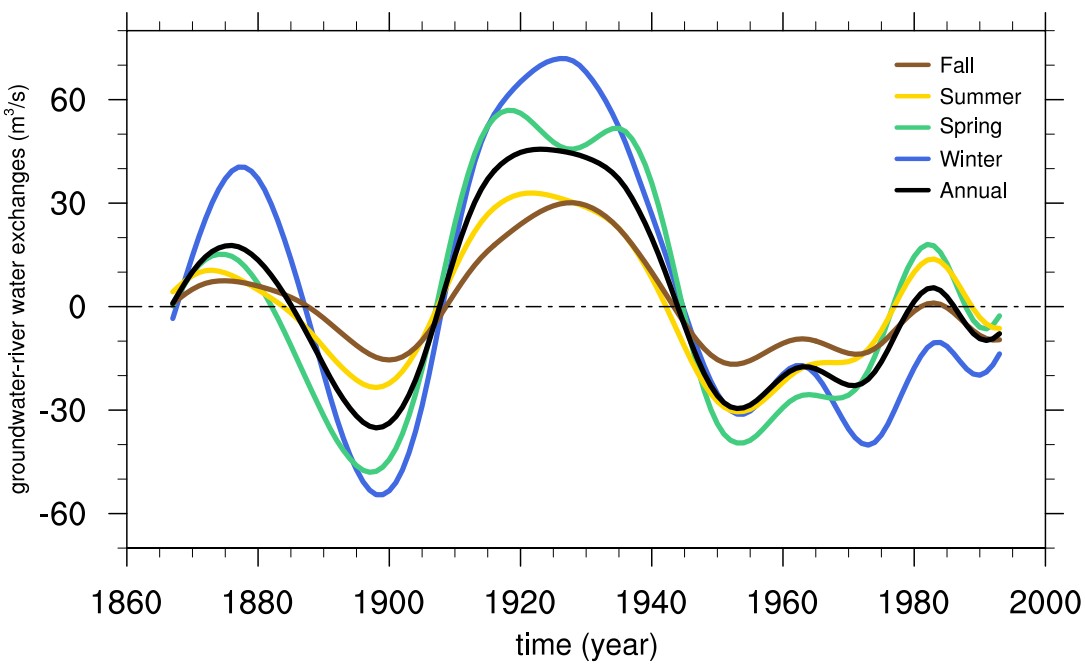

**Figure 9.** Annual low-pass filtered anomalies of the (black) annual, (blue) winter, (green) spring, (yellow) summer, (brown) autumn average groundwater-river water exchanges over the Seine basin in the reconstruction. The reference period is 1852-2008.

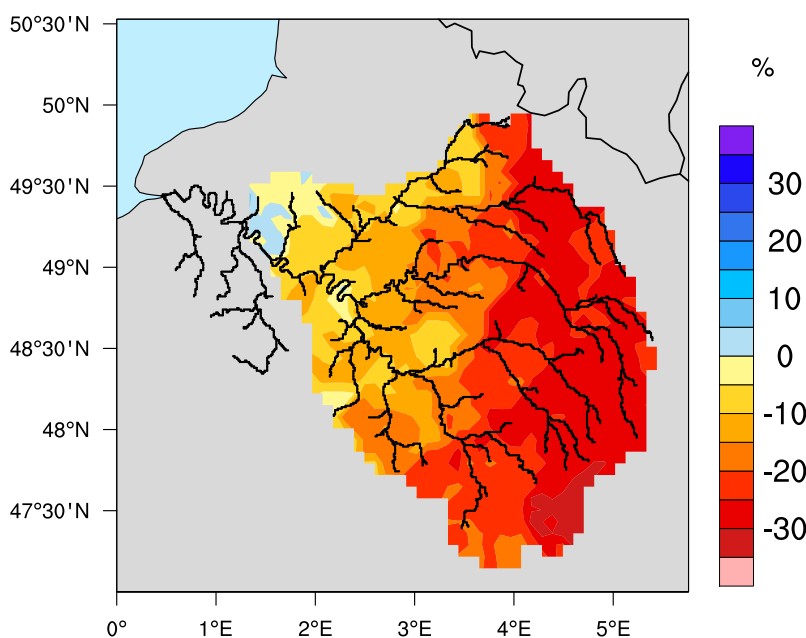

**Figure 10.** Relative differences (%) in the ratio between the total runoff and precipitation of the reconstruction, calculated in summer between the negative multidecadal phases of the Seine river flows (1885-1905 and 1940-1960) and the positive phases (1910-1930 and 1975-1995). The reference for the calculation of the relative anomalies is the average on these four periods. Dots show where the differences are significant based on a Student's t-test (p-value< 0.05)

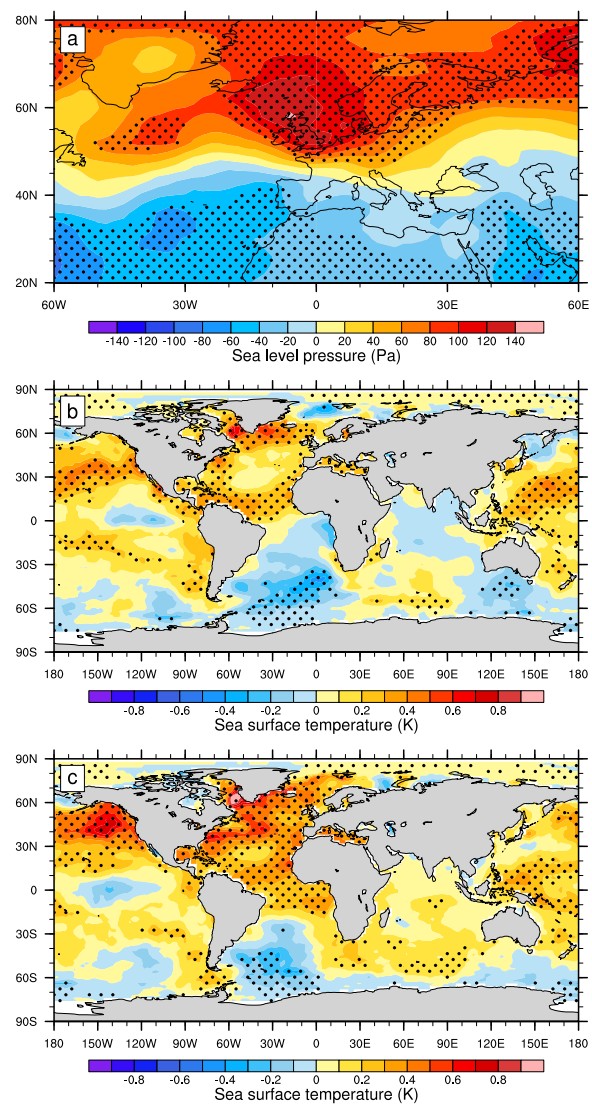

**Figure 11.** Differences in (a) detrended pressure at sea level (SLP) (Pa), (b) detrended sea surface temperature (SST) (K) between the negative multidecadal phases of the Seine river flows (1885-1905 and 1940-1960) and the positive phases (1910-1930 and 1975-1995). (c): same as (b) with a negative lag of 10 years. SLP comes from the NOAA 20CRv2c reanalysis. SST comes from ERSSTv5 (Huang et al., 2017). The trends are calculated with the EEMD method and subtracted at each grid point. Dots indicate where the differences are significant based on a Student's t-test with a p-value < 0.05.

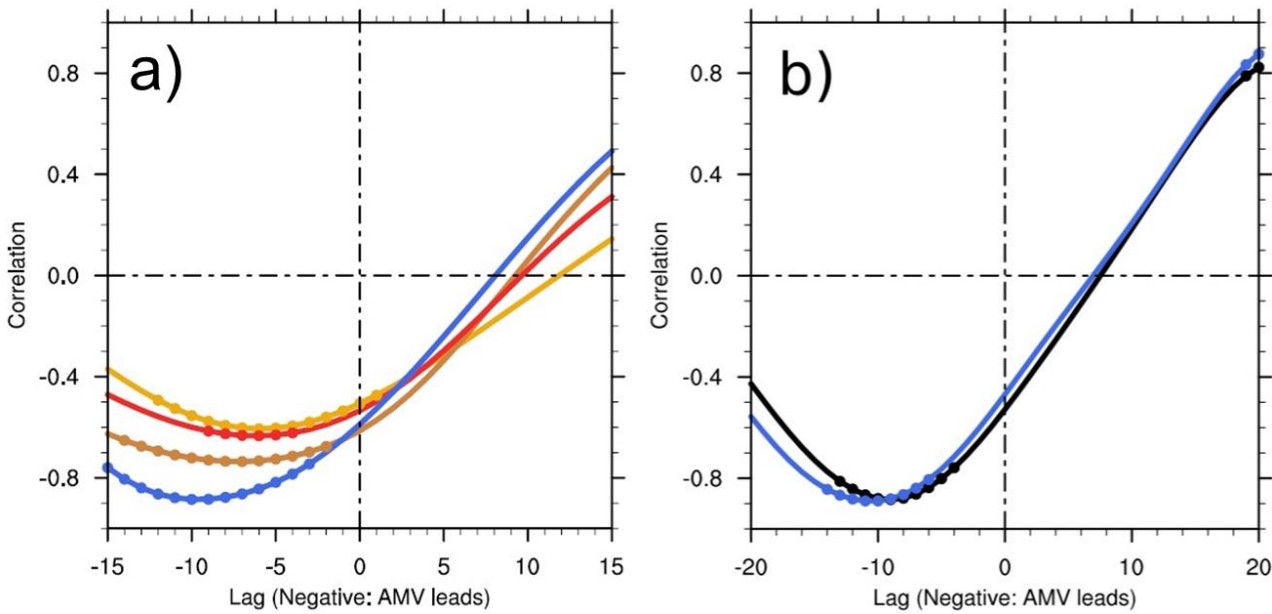

**Figure 12.** (a) Lagged correlations (lag in year, negative values: AMV leads) between the paleoclimate AMV index of Wang et al. (2017) and the long series of spring precipitation observed at Paris (Slonosky, 2002) over the period 1780-1889 (yellow), 1890-1989 (brown) and 1779-1989 (red). In blue, the correlations between the AMV index calculated from the 20CRv2c reanalysis and the spring precipitation from the reconstruction over the period 1882-1979. (b) Lagged correlations (lag in year) between the AMV index calculated from the 20CRv2c reanalysis and the spring river flows from (blue) the reconstruction at Paris and (black) the observations at Paris Austerlitz over the period 1882-1979. The trends in SST, precipitation and river flows series are calculated with the EEMD algorithm and subtracted from the original series before the calculation of the correlations. The series are filtered with a Lanczos filter with a 21-year window. The points represent significant correlations with a p-value < 0.05 (at least 95% of the null hypothesis is rejected) according to the "phase-scrambled bootstrapping test" (Davison and Hinkley, 1997)

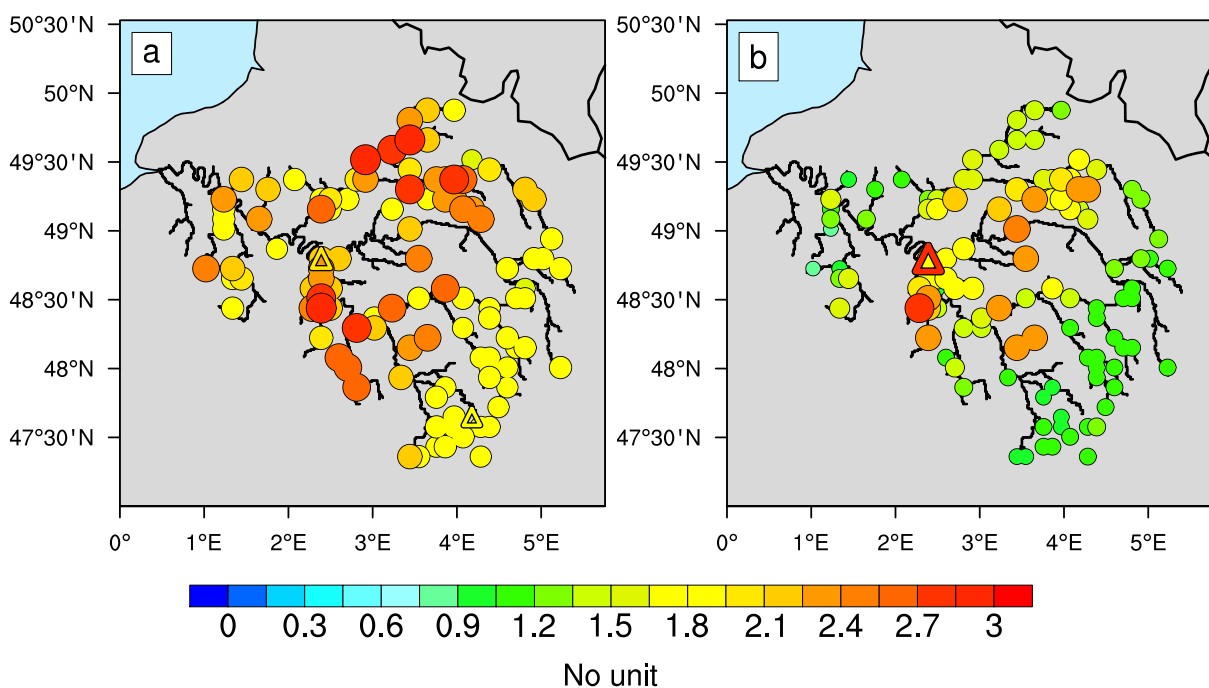

**Figure 13.** (a) Ratio of the number of days with river flow equal to or greater than the 95th percentile between the positive multidecadal phases of the Seine river flows (1910-1930 and 1975-1995) and the negative phases (1885-1905 and 1940-1960) calculated for the reconstruction (dots), and for the two available long term observation series (triangles, see figure 1). Only the days of the flood season are used (November to April). (b) same as (a) for the days with river flows equal to or lower than the 5th percentile between the negative and the positive multidecadal phases, considering the whole year.

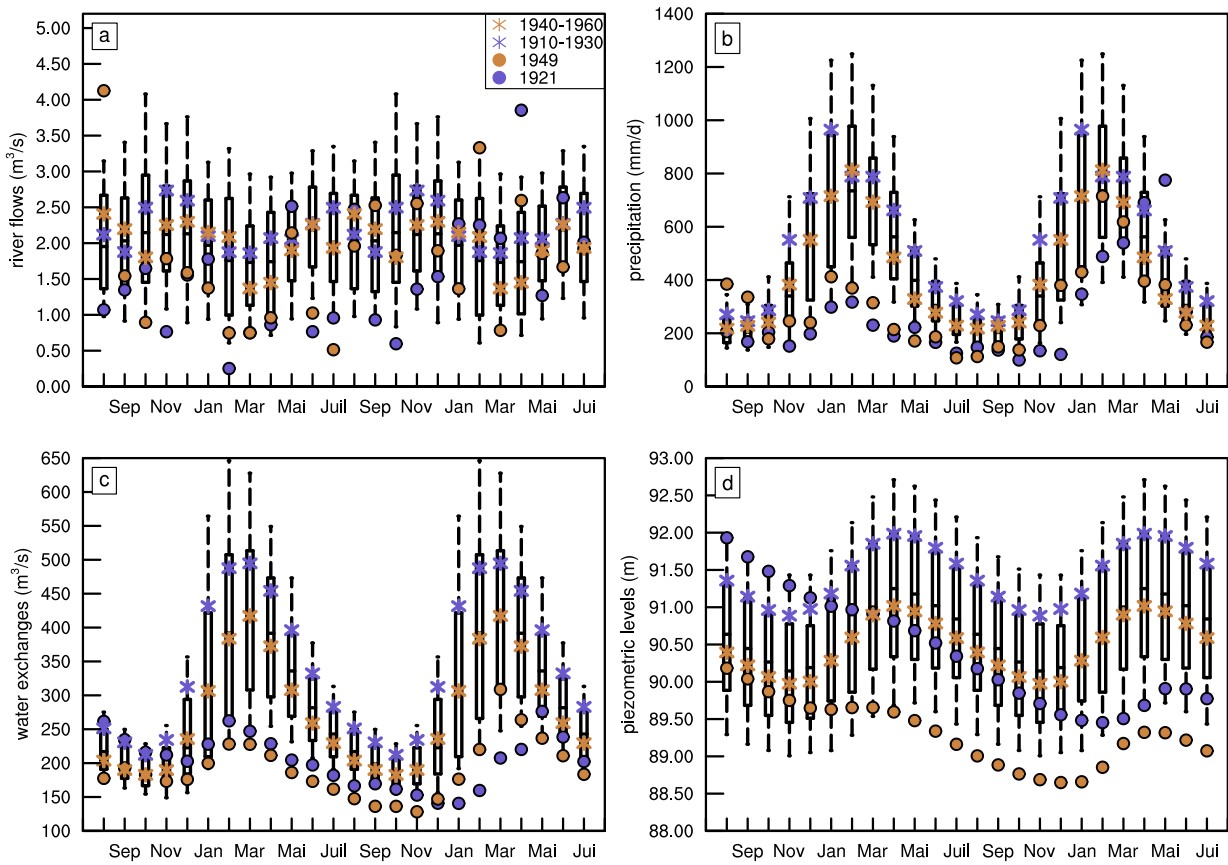

**Figure 14.** a) Evolution of monthly precipitation (mm/d) averaged over the Seine basin from august 1920 to July 1922 (purple dots) and from August 1948 to July 1950 (beige dots). The boxplots represent the climatological monthly distributions calculated over the 1852-2008 period. Purple (beige) crosses show the monthly average over the 1910-1930 (1940-1960 period). (b) Same as (a) for river flows at Poses (m3/s), (c) water exchanges between groundwater and rivers (m3/s) and (d) the piezometric levels (m) of the unconfined part of the aquifer. The data come from the Seine reconstruction. The boxplots are defined with the 10th percentile, 25th percentile, median, 75th percentile and 90th percentile.