# Peer review of "Influence of multidecadal variability on high and low flows: the case of the Seine basin"

_Hydrology and Earth System Sciences, 2019_

## Referee Comment (RC1) · Anonymous Referee #1 · 26 Aug 2019

The authors present a hydrometeorological reconstruction established using a combination between a statistical downscaling method and data assimilation. This reconstruction is then used to study the multidecadal hydroclimate variability of the Seine basin, as well as the influence of the multidecadal variations on extreme events. The paper is well constructed and address relevant questions about the mechanisms of hydrological variability on the Seine basin. Figures are clear, and results and conclusion are relevant. I mostly have questions about some details in the methodological part.

**Comments that should be addressed in the paper**

**Introduction**

p.2 l.22-32 – After reading this paragraph, we think that statistical downscaling method are really not appropriate and that these methods should be discard. But in fact, you use this method with a second step combining observations with the results of a statistical downscaling method to improve them. Maybe it would be worth to reformulate some sentences of this paragraph to explain that these methods are not enough to well characterize climate variability, and that we should use an 'add on' or an 'evolution' to take into account observations. This couldn't be considered as a completely different method.

p.2 l.33 to p.3 l.3 – Indeed, using observations could improve the results of a statistical downscaling method. But observations are also inhomogeneous so at the end, isn't it the same disadvantage than statistical downscaling methods? Inhomogeneous trends? Combining observations does not cancel the disadvantages of the downscaling (so we have inhomogeneous trend from the reanalysis + inhomogeneous trend from the observations?).

Why only talking about statistical downscaling methods? What other methods could be used to reconstruct the past (dynamical downscaling, weather generators, …) and why is it better to choose statistical downscaling + data assimilation?

p.2 l.6 Why the Seine basin? Can you add some explanations?

P2 l.8 Extending until 1850 leads to using 20CR between 1850 and 1900 (quality - and very large dispersion over the 56 members) + using very few observations (so inhomogeneous trends). How can we drive conclusions over this period given the poor data quality?

p.2 l.16 – Are the observations independent for the evaluation?

**Data, models and methods**

p.4 l.3 – The SMR developed by Moisselin, 2002, show significant inhomogeneous trends. It could be worth to add this fact the in text.

p.5 l.1-2 – Is Safran really independent from the reconstruction as the same observations are certainly used in Safran and data assimilation?

p.5 l.16 – At this point two questions:

- What about the spatial and temporal coherence? I suppose the spatial is respected as the same domain is used for the entire basin, but the temporal one?

- Isn't it possible, for each day, to constrain the results of the downscaling to create trajectories in function of observations, instead of creating all the 56 trajectories independently and them choose the closest to the observations? In the first case, the final trajectories would be different than the trajectories from the downscaling method without data assimilation. -> I believe this is done by the

process explained l.17-32, and with only 3 trajectories instead of 56. Maybe reverse the explanations in the text, explaining the daily constrain before talking about the monthly one (not mention the monthly one before).

p.5 l.25 – Indeed, all the variables from Safran are obtained but the predictors for the downscaling method are only optimized for precipitation and temperature, isn't it? So what about the quality of the reconstruction for other variables, did you assess it? Is it good enough to used them in the hydrological model? It is for example possible to compare the reconstructed signals to Safran on the recent period.

p.6 l.7 – Is it possible to give a little more details about these tests? Comparison to what?

p.6 l.8 – Why 3 analogue days?

p.6 l. 10 – Is it possible to sum up the method in Bonnet, 2017 in a few lines?

p.6 l.15 – What is the influence of using different types of observations for the data assimilation on different periods?

p.6 l.22 – Is the spatial (temporal?) coherence conserved after the correction of these biases?

p.6 l.22 – Maybe change the title "Method" in "Extraction of multidecadal variability" as the previous paragraph was also talking about methodological facts?

**Evaluation of the Seine reconstruction**

p.7 l.20 – On Figure 3, we see that before 1900, reconstructions are not close to observations. This is not a surprise as 20CR has a poor quality before 1900 and the network of observations is less dense. The signal at Paris seems more "flat" than observations, with under and over estimations.

p.8 l.4 Which type of correlation is used?

**Multidecadal hydroclimate variations**

p.9 – Isn't it difficult to drive conclusions about other variables than P, T or Q as they are difficult to model in hydrological models?

**Conclusion**

Wouldn't it be more logical to reverse the conclusion part (for now, first) and the discussion part (second) in the conclusion?

**Orthographic corrections**

P.1 l.4 – Reformulate the sentence "This method improves the representation of daily flow as well as at longer time step"

P.1 l.4 – Provides

P.1 l.8 – Maybe "regulate" instead of "modulate" would be better?

p.1 l.12 – to influence the drought intensities

p.1 l.16 – Missing a "," after "for example"

p.1 l.22, p.2 l.11 – Same remark for modulate / regulate

p.2 l.1 – Verify the expression "internal variability in climate and/or…"

p.2 l.20 – "It" not necessary in "which makes …"

p.2 l.25 – "." In the middle of the sentence, between "downscaling" and "of"

p.6 l.2 – Maybe add "spatial" before "error". The sentence is not really clear, maybe there is a way to rephrase it, talking about spatial errors for both precipitation and temperature.

p.12 l.27 – The SSTs  are

---

## Referee Comment (RC2) · Anonymous Referee #2 · 13 Sep 2019

This paper details research conducted on the link between multidecadal hydroclimate variations and streamflow in the Seine basin in France. Unfortunately, I have found it difficult to determine the scientific contribution of this paper due to its flaws in language, layout and lack of detail in the methods. I am recommending that this paper be significantly revised, in order that we may better understand the outcomes of this research. The major points that I recommend are:

1. The paper's motivation needs to be better set out. The abstract states that precipitation and groundwater modulate river flows, and that extreme events are influenced by hydroclimate variations. This is not news to the hydrological community. What is novel here? Why is this interesting, and to who, and why?

2. You are missing significant volumes of literature surrounding connections between atmospheric circulation patterns and streamflow across wider Europe. e.g. (among many others)

   - Folland, C. K., Hannaford, J., Bloomfield, J. P., Kendon, M., Svensson, C., Marchant, B. P., Prior, J., and Wallace, E.: Multi-annual droughts in the English Lowlands: a review of their characteristics and climate drivers in the winter half-year, Hydrol. Earth Syst. Sci., 19, 2353-2375, https://doi.org/10.5194/hess-19-2353-2015, 2015.
   - Eva Steirou, Lars Gerlitz, Heiko Apel, Bruno Merz, Links between large-scale circulation patterns and streamflow in Central Europe: A review https://doi.org/10.1016/j.jhydrol.2017.04.003

3. You need to explain your method much more clearly. Can you summarize the method in Bonet et al (2017) as it is mentioned over and over, and is seemingly critical to the understanding of your method here.

4. The paper's language be reviewed by a confident English speaker. The sentence structures are often incorrect, and the tenses are jumbled, which has made the paper very difficult to read. The English appears to improve later in the paper, so I think a little more effort is required.

5. The papers structure and headings need to be amended, especially in the methods section.

6. Most of your figures have no legends, and many do not have appropriately descriptive axis labels. Please correct this to aid interpretation

7. Nearly all of your figures rely on red/green differentiation. 1 in 10 of the male readers will not be able to see this due to common colour-blindness. Please amend your color schemes to avoid this issue.

I have felt unable to go into the detail of the research for these reasons, and would be happy to re-review the paper once these issues have been addressed.

---

## Referee Comment (RC3) · Anonymous Referee #3 · 16 Sep 2019

Influence of multidecadal hydroclimate variations on hydrological extremes: the case of the Seine basin Bonnet et al.

Overall this is an interesting paper that develops reconstructed flows for the Seine basin back to the 1850s. Variability in the reconstructions is then assessed and linked to SSTs. While this work will be of interest to the readers of the journal I do not think it is ready for publication. It is difficult to read at present and clarification is needed at times as to what is done. A fuller discussion of the limitations, uncertainties and assumptions of the work is needed. Discussion of other modes of climate variability that could be influential would also be welcome.

I highlight some issues below which the authors should address. I hope that these are seen as constructive, as I believe this could be a nice addition to the literature.

[Figure]

Title – hyphen multi-decadal

In addition hydro appears twice in the title, while the paper really only deals with SSTs as a source of variability . Suggest title change to "Influence of multi-decadal variability on hydrological extremes: the case of the Seine basin. You could include SST before variability if you wish.

Overall the manuscript needs to be thoroughly edited. There are lots of problems with the use of plurals throughout eg. Precipitations , which should always be singular.

It is not clear to the reader what you mean by multi-decadal phases. This appears throughout the paper and I strongly suggest you define use of it early on or use a different term. I assume it means wetter than average or drier than average periods.

The abstract states the obvious a little too much, eg. Wet periods are conducive to more flooding, dry periods to droughts. I would be more interested to learn what specifically your reconstructions offer and the new insights they provide.

In addition, the final sentence of the abstract gives the impression that this paper looks at dynamical chains in how ssts influence atmospheric circulation. It does not.

The literature review mentions the AMV as an important source of variability, but what about others modes of variability eg. The NAO, East Atlantic Pattern etc. There is more than just the AMV at play and it is my view that this should be reflected in the literature review. These other modes of variability could also be examined for their relationships with the extremes in the reconstructed flows or discounted if the literature has previously addressed these issues.

In your critique of 20CR on page 2 line 30 you explicitly state that that "this approach is far from optimal, as it does not make use of the long-term meteorological observations that may exist". I would strongly recommend a citation for this or remove.

There are uncertainties not considered in this paper related to input data, hydrological modelling, other datasets used etc. It would be welcome if the discussion could, in a

paragraph, flesh these out.

As mentioned, the methodology is complex and difficult to follow. The use of analogues to derive daily precipitation seems to move from 2800 samples to 60 to 3. Why three, why such a decrease in sample size and are you left with a large enough sample?

Similarly there are assumptions made that things like channel dimensions and conveyance capacity remain stationary over time. Recent research has shown that they also react to variability Slater, L.J., Khouakhi, A. and Wilby, R.L., 2019. River channel conveyance capacity adjusts to modes of climate variability. Scientific reports, 9(1), pp.1-10.

In terms of your methods does your use of analogs constrain the variability to that of the shorter record? The methodology is complex and hard to follow as it is written. Previously published work on which this data is based needs to be shortly summarised.

In addition it would be helpful to the reader to provide a flow diagram of the key steps in the workflow. For example, I have read the paper multiple times and at the start of the results I am unsure what the reference simulation is.

I don't like the word anthropized – heavily impacted would work just as well.

It appears that the reconstructions persistently underestimate the annual maxima (fig 4). What does this say about your reconstruction, does it affect your results and how might this be improved?

Have the piezometic level data been quality assured?

It is stated on page 7 line 29 that "The length of the reconstruction allows to show that multidecadal variations are also present before the 20th century….." Is this not expected. In the same sentence what do you mean by negative phase.

Throughout you could quantify these above and below average phases in terms of the magnitude of their anomaly.

Indeed what can your work tell us about an appropriate baseline for assessing changes in the Seine flows – one that accounts for the types of variability you see. Thirty year baselines often only sample one component of mult-decadal variations.

There are a number of statements that require clarification or a more precise wording, such as: multi-decadal phases? P7 line 10 over what period are correlations derived? Page 8 line 23 A partially captive part of the aquifer is not represented in the model – what is a partially captive part? Pg 11 line 11 a large number of stations have a ratio between one and two….with a ratio of 1.1 consistent with the reconstruction in that region. I cant interpret this. Pg 11, line 24 strong positive multidecadal phase?

Is it fair to call a day with flow above the 95th percentile a flood day. Is it fairer to say a high flow day? The same issue applies to drought days, are these more fairly described as low flow days? In reality you are not strictly looking at floods and droughts in this part of the analysis.

It would be useful to outline a potential or plausible chain of causation as to how SSTs in the North pacific relate to Seine flows.

Do your extended flows add additional insight into AMV related variations in flows than done in previous work. Has the magnitude of anomalous periods been the same or different in your reconstructions relative to previous work?

Section 4.1 is hard to follow, it might be worthwhile organising by season. At times the text jumps from season to season

I think section 6 needs to come earlier in the paper, given it importance to the message of the paper.

---

## Author Comment (AC1) · 29 Oct 2019

**Response to reviewer#1 of: "Influence of the multidecadal hydroclimate variations on hydrological extremes: the case of the Seine basin" by R. Bonnet, J. Boé and F. Habets.**

First, we would like to thank the reviewer for his carefully reading, interest in our study and the insightful comments that helped us to improve the manuscript. Some points are shared with the other reviewers, especially on the description of the method used to develop the hydrometeorological reconstruction, which was not clear enough. We made major modifications to the description of the method, which is now in a specific section. A diagram is now added to bring more clarity. The introduction was also greatly improved. We also added a new figure in the section about the "Role of large-scale circulation and influence of ocean variability", which improves and completes the insights of this section (now the 6th section). Please find below the answers to the comments point-by-point. For clarity, all reviewer comments are in **bold**.

**Summary:**
**The authors present a hydrometeorological reconstruction established using a combination between a statistical downscaling method and data assimilation. This reconstruction is then used to study the multidecadal hydroclimate variability of the Seine basin, as well as the influence of the multidecadal variations on extreme events. The paper is well constructed and address relevant questions about the mechanisms of hydrological variability on the Seine basin. Figures are clear, and results and conclusion are relevant. I mostly have questions about some details in the methodological part.**

**Introduction:**
**p.2 l.22-32 – After reading this paragraph, we think that statistical downscaling method are really not appropriate and that these methods should be discard. But in fact, you use this method with a second step combining observations with the results of a statistical downscaling method to improve them. Maybe it would be worth to reformulate some sentences of this paragraph to explain that these methods are not enough to well characterize climate variability, and that we should use an 'add on' or an 'evolution' to take into account observations. This couldn't be considered as a completely different method.**

The third reviewer also highlighted the lack of clarity of this paragraph. Please, find below the new version of the paragraph added in the revised manuscript:

"To move forward, long-term hydrometeorological reconstructions based on hydrological modelling have been developed (e.g Kuentz et al. 2015; Caillouet et al. 2016). Due to the scarcity of meteorological observations in the early 20th century (Minvielle et al., 2015), the meteorological forcing needed for hydrological modelling must first be reconstructed. The recent release of long-term global atmospheric reanalyses (e.g. Twentieth Century Reanalysis (20CR,Compo et al., 2011) from the National Oceanic and Atmospheric Administration (NOAA)) opens great opportunities in that context. Statistical downscaling methods, typically used in climate change impact studies, can be applied to derive the high resolution meteorological forcing necessary for hydrological modelling from these global atmospheric reanalyses, as in Caillouet et al. (2016). This approach presents two main limitations. First, the quality of the reconstruction depends on the quality of the reanalyses. As the density of assimilated observations (e.g. surface pressure in NOAA 20CR, Compo et al., 2011) strongly evolves over time, potential unrealistic trends and/or low frequency variations may exist (Krueger et al., 2013; Oliver, 2016; Bonnet et al., 2017). Second, this approach does not take advantage of the long-term local meteorological observations that may exist.

Given these limitations, following the same general idea as Kuentz et al. (2015), Bonnet et al. (2017) presented a new hybrid method that combines available long-term monthly observations of precipitation and temperature with the results of a statistical downscaling method applied to long-term atmospheric reanalyses. Compared to standard dynamical or statistical downscaling methods that only use large scale information and do not take advantage of local observations (e.g. temperature and precipitation) a more realistic representation of local hydroclimate variations may be obtained (Bonnet et al., 2017)"

**p.2 l.33 to p.3 l.3 – Indeed, using observations could improve the results of a statistical downscaling method. But observations are also inhomogeneous so at the end, isn't it the same disadvantage than statistical downscaling methods? Inhomogeneous trends? Combining observations does not cancel the disadvantages of the downscaling (so we have inhomogeneous trend from the reanalysis + inhomogeneous trend from the observations?).**

Indeed, uncertainties are also present in the observed precipitation and temperature series used to constrain the results of the statistical downscaling. They are however still interesting to use as they provide local information that are not given by large-scale reanalyses. The third reviewer also pointed out that a discussion on the different uncertainties associated with the reconstruction method was missing. We added a discussion about this in the conclusion, including your comments on the observation. Please find below the paragraph:

"Although the reconstruction developed in this study is an interesting tool for studying the past variability of the hydrological cycle over the Seine basin, it is obviously not perfect. Uncertainties are present throughout the modelling chain. The statistical downscaling method used at the first step of the reconstruction method assumes that the learning period, over which the large-scale reanalysis and the Safran analysis overlap (1959-2010), is representative of the meteorological conditions of the 1851-2010 period. The consistent performances of the reconstruction over the entire period shown by several analyses in this study suggest that this hypothesis has no major impact on our results. Important uncertainties are associated with the 20CRv2c reanalysis at the beginning of the period, due to the smaller number of assimilated observations (Krueger et al., 2013). We use monthly homogenized local precipitation and temperature observations to constrain the results of statistical downscaling in order to improve the temporal homogeneity of the reconstruction, but the homogenization method is not state-of-the-art. The good agreement between the low-frequency variations of the homogenized monthly precipitation series and of the Global Precipitation Climatology Centre dataset (Schneider et al., 2008) from 1901 to 2011 (not shown) still gives good confidence in the overall realism of the multidecadal variations described in this study."

**Why only talking about statistical downscaling methods? What other methods could be used to reconstruct the past (dynamical downscaling, weather generators, ...) and why is it better to choose statistical downscaling + data assimilation?**

We think that using only dynamical downscaling or weather generators shares the same weaknesses associated with statistical downscaling in that context: only large scale information is used. We now make it clear in the revised manuscript:

"Given these limitations, following the same general idea as Kuentz et al. (2015), Bonnet et al. (2017) presented a new hybrid method that combines available long-term monthly observations of precipitation and temperature with the results of a statistical downscaling method applied to long-term atmospheric reanalyses. Compared to standard dynamical or

statistical downscaling methods that only use large scale information and do not take advantage of local observations (e.g. temperature and precipitation) a more realistic representation of local hydroclimate variations can be obtained (Bonnet et al., 2017).

**p.3 l.6 Why the Seine basin? Can you add some explanations?**

We added a point to justify the choice of the Seine basin in the new version of the introduction. Please find below the explanations of this choice:

"Focusing on the Seine basin (Figure 1), one of the main French river basins, we are able to extend the reconstruction back to the 1850s. A major interest of this basin is indeed the existence of a few long and varied series of observations, which are useful either to develop or evaluate the reconstruction method"

**P3 l.8 Extending until 1850 leads to using 20CR between 1850 and 1900 (quality - and very large dispersion over the 56 members) + using very few observations (so inhomogeneous trends). How can we drive conclusions over this period given the poor data quality?**

This period has indeed to be interpret with caution. In our case, as the monthly observations used in the study are homogenized and as we considered only stations with no missing values, it is more likely that adding this information improve the quality of our reconstruction. Additionally, the combination of large scale atmospheric reanalysis and long-term observations has shown promising results in previous studies (Kuentz et al., 2015, Bonnet et al., 2017). Even if there are a lot of uncertainties between the years 1850 and 1900, the use of different kind of dataset allows to derive some common conclusion. As said 3 points before, we added a discussion about the uncertainties related to the reconstruction in the conclusion section.

**p.3 l.16 – Are the observations independent for the evaluation?**

All the hydrological variables except precipitation used in the evaluation (river flows, aquifer levels) are independent of the hydrometeorological reanalysis. This is why we focused the evaluation on these variables. Only observation of precipitation and temperature are combined with the results of the statistical downscaling method in the reconstruction method and are, therefore, not independent from the hydrometeorological reconstruction. We added this clarification in the revised version of the manuscript (below).

"These observations, which are used in particular to evaluate the hydrometeorological reconstruction developed in this study, are independent of it."

**Data, models and methods**

**p.4 l.3 – The SMR developed by Moisselin, 2002, show significant inhomogeneous trends. It could be worth to add this fact the in text.**

We are not aware of studies that demonstrate inhomogeneous trends in the SMR developed by Moisselin (2002). But it is true that the method is no longer state-of-the-art, and we now acknowledge it in the manuscript. Also, as now said in the discussion about uncertainties in the conclusion (your second comment), we made a comparison of low-frequency variations in precipitation between the SMR and the GPCC data, and both dataset show a good agreement over France. As we look at multi-decadal variability and we don't focus on long-term trends, this is not an important issue in our study. Note that in order to limit the potential

influence of missing values on reconstructed long-term trends, we only use SMR stations with no missing values over the 1885-2005 period of interest.

**p.5 l.1-2 – Is Safran really independent from the reconstruction as the same observations are certainly used in Safran and data assimilation?**

Safran is not independent from the reconstruction. Some observation stations of precipitation and temperature used to constrain the statistical downscaling are also used in the Safran analysis. However, the only difference between the Safran-Surfex-AquiFR simulation and our hydrometeorological reconstruction is the quality of the meteorological reconstruction, as they share the same hydrological model. We clarified this point in the new version of the manuscript:

"A simulation based on the Safran-Surfex-AquiFR system is available over the 1958-present period. This so-called reference simulation in the following is used for the evaluation of the hydrometeorological on their common period. As they share the same hydrological model, potential differences between the reconstruction and the reference simulation only depend on the quality of the reconstructed meteorological forcing."

**p.5 l.16 – At this point two questions:**
**- What about the spatial and temporal coherence? I suppose the spatial is respected as the same domain is used for the entire basin, but the temporal one?**

The spatial coherence is respected as all the meteorological variables come from the same analogue day. As the analog days are selected from the same atmospheric reanalysis, a temporal coherence is also present in the meteorological reconstruction. We added a paragraph on this point in the new section on the development of the Seine reconstruction in the revised manuscript. Note also that evaluating river flows is an indirect way to assess that the spatial and temporal coherence are correct: without a good representation of spatial and temporal coherence, the simulated river flows would not be realistic.

"This approach benefits from the advantages of the analog statistical downscaling method. From the analog days, all the meteorological variables from Safran necessary to force the Surfex-AquiFR hydrological model were obtained. The spatial and inter-variable consistencies were maintained after this procedure, because for each day of the reconstruction the entire map of precipitation (and temperature, humidity etc.) over France from Safran was selected based on a single analog day."

**- Isn't it possible, for each day, to constrain the results of the downscaling to create trajectories in function of observations, instead of creating all the 56 trajectories independently and them choose the closest to the observations? In the first case, the final trajectories would be different than the trajectories from the downscaling method without data assimilation. -> I believe this is done by the process explained l.17-32, and with only 3 trajectories instead of 56. Maybe reverse the explanations in the text, explaining the daily constrain before talking about the monthly one (not mention the monthly one before).**

Yes, it is possible in theory. However, to be effective, the daily constraint needs a large sample of analogue days to find a good match with observed temperature and precipitation. But in practice the quality of analogue days rapidly decreases: with the limited sample sizes allowed by observations, it is impossible to find for example 100 very good analogue days (i.e. with large scale predictors close to the target). The interest here to downscale the 56 members of the reanalysis is to create an ensemble of possibilities large enough while maintaining the quality of these analogue days. As mentioned by the other reviewers, this

section was not clear and difficult to follow. We made important modifications in this section (now a particular section), which is now much clearer. A description of the method used in Bonnet et al., 2017 was added (as asked by the two other reviewers). We also added a diagram to make this section easier to follow, as suggested by the third reviewer. Please find this new section below:

[revised manuscript text omitted]

**p.5 l.25 – Indeed, all the variables from Safran are obtained but the predictors for the downscaling method are only optimized for precipitation and temperature, isn't it? So what about the quality of the reconstruction for other variables, did you assess it? Is it good enough to used them in the hydrological model? It is for example possible to compare the reconstructed signals to Safran on the recent period.**

With our hybrid approach, the quality of other variables (specific humidity, wind etc.) is not worse than the direct results we could obtain with statistical downscaling alone. As predictor for the statistical downscaling part of the method, we use 4 variables: precipitation, temperature, specific humidity at 850hPa and the pressure at sea level. With these predictors, we suppose that we have a good representation of the large-scale atmospheric conditions which influence the meteorological conditions over the Seine basin. Then, the reconstruction is optimized for temperature and precipitation by constraining them with the available observations. Several sets of predictors were tested with Safran for variables of interest such as the precipitation, the temperature, but also wind speed, relative humidity and solar radiation. We don't show that in the article to keep it short. Note that the evaluation of river flows and aquifer levels indirectly shows (with independant data) that the representation of all forcing variables is "good enough" to reproduce river flows reasonably for our purposes.

**p.6 l.7 – Is it possible to give a little more details about these tests? Comparison to what?**

A methodological choice had to be done here about the weight given to each of the two variables. We chose to give a little more weight to precipitation, which is an essential variable in the representation of river flows. A test carried out on the impact of the weight given to the temperature variable is illustrated Figure R1 below.

Adding temperature to the daily constraint greatly improves the representation of temperature with Safran compared to the daily constraint based on precipitation only, with a correlation gain of about 0.2 at the daily time scale and 0.07 at the monthly time scale. However, with a greater weight for temperature the correlations for precipitation logically slightly decrease.

A balance must therefore be found to improve the daily correlations of temperature without affecting too much the daily correlations of precipitation, for which a good representation is essential to study the high and low flows of the Seine basin, as well as extreme hydrological events. A weight of 0.5 is finally assigned to the temperature variable for the daily constraint.

We added a paragraph to illustrate some tests realized during the development of the reconstruction in the new section:

"Multiple tests have been conducted to set-up the different ad-hoc aspects of the method, trying to obtain the best overall hydrometeorological reconstruction. These tests concern, for example, the best combination of weights given to precipitation and temperature errors, the number of analogs selected at each steps etc. For example, selecting only the 3 best analog days leads to best overall performance in capturing daily and monthly variations. Using more analog days may allow for a better representation of monthly variations but degrade the representation of daily variations."

[Figure]

*Figure R1: Spatial distribution of the correlations between (a-c) daily and (b-d) monthly (a-b) precipitation and (c-d) temperature over the Seine basin. Correlations are calculated between the Safran analysis (considered as observations) and the precipitation and temperature derived from the statistical downscaling method only constrained at daily time scale by (blue) precipitation only and (black) precipitation and temperature considering different weights for the temperature (indicated in X axis). The correlations are calculated on the 1958-2005 period and the series have been deseasonalyzed beforehand. The boxplots show the minimum/25th percentile/median/75th percentile and the maximum.*

**p.6 l.8 – Why 3 analogue days?**

An important objective of our hydrometeorological reconstruction is to improve the representation of daily river flows. As a result, a balance has to be found for the number of analogue days to be used. The greater the number of analogue days is, the farther some analogue days are from the target day, with likely in the end a degradation of the representation of precipitation and temperature, and therefore of river flows. On the other hand, too few analogue days could limit the improvement in low frequency variations expected from the monthly constraint (as there is less spread to find a good monthly

trajectory with fewer analogue days). We therefore made different tests in order to find the best number of analogues to retain at the different steps.

This is illustrated in figure R2, which shows the results of one of these tests. It shows that the daily correlations between reconstructed and observed temperature and precipitation decrease when the number of analogue days increases (Figure 1a and c). After testing different possibilities, we decided to keep the 3 best analogue days from the daily constraint. These 3 analogue days are then used to apply the monthly constraint. With 3 analogues, the ensemble is large enough for the monthly constraint to be effective.

Figure R2 also shows that the double constraint method, at daily and then monthly time scales, greatly improves the daily correlations of precipitation and temperature compared to the statistical downscaling method alone, or to the downscaling method only constrained at monthly time scale (Figure 4.11 a and c).

[Figure]

Figure R2: Spatial distribution of the correlations between (a-c) daily and (b-d) monthly (a-b) precipitation and (c-d) temperature over the Seine basin. Correlations are calculated between the Safran analysis (considered as observations) and the precipitation and temperature derived from (brown) the reconstruction developed in Bonnet et al., 2017, (green) the downscaling method alone, (purple) the downscaling method only constrained by monthly precipitation and temperature, (blue) the downscaling method only constrained by daily precipitation and (red) the downscaling method constraint by daily and monthly precipitation and temperature, based on different tests for the number of analogs used for the monthly constraint (X axis). The correlations are calculated on the 1958-2005 period and the series have been deseasonalyzed beforehand. The boxplots show the minimum/25th percentile/median/75th percentile and the maximum.

**p.6 l. 10 – Is it possible to sum up the method in Bonnet, 2017 in a few lines?**

We added a summary of the method used by Bonnet et al., 2017 in the new section about the development of the hydrometeorological reconstruction. Please find the description of this method four points above.

**p.6 l.15 – What is the influence of using different types of observations for the data assimilation on different periods?**

The quality of our reconstruction may be reduced for periods constrained only by one monthly series of precipitation (1850-1885 or the 2005-2010) in comparison to the 1885-2005 period, constrained by daily and monthly precipitation and temperature. The results, therefore, have to be interpreted with caution on these periods. We added a sentence about this point at the end of the new method section:

"To sum up, the hydrometeorological reconstruction developed on the Seine basin is constrained on a daily basis over the period 1885-2003 by observations of precipitation and temperature (SQR), on a monthly basis over the period 1885-2005 by homogenized observations of precipitation and temperature (SMR), and over the 1852-1884 and 2005-2008 periods by the monthly series of precipitation at Paris (Slonosky, 2002) (see section 2.1 for more details). The results, especially at the daily time scale have therefore to be interpreted with more caution over the period only constrained by the monthly series of precipitation."

**p.6 l.22 – Is the spatial (temporal?) coherence conserved after the correction of these biases?**

As these are only climatological biases, this doesn't influence the coherence of the reconstruction.

**p.6 l.22 – Maybe change the title "Method" in "Extraction of multidecadal variability" as the previous paragraph was also talking about methodological facts?**

A large part of this section is about the way we extracted the multidecadal variability, but it is also about the way we deseasonalized the series before calculating the daily and monthly correlation as well as the acronyms used for the seasons. We therefore preferred to keep "Method" as title.

**Evaluation of the Seine reconstruction**
**p.7 l.20 – On Figure 3, we see that before 1900, reconstructions are not close to observations. This is not a surprise as 20CR has a poor quality before 1900 and the network of observations is less dense. The signal at Paris seems more "flat" than observations, with under and over estimations.**

We think that it is difficult to disentangle the respective impacts of errors in the reconstruction (due to 20CR or the low-density of the resolution network) and potential non anthropogenic influences, or measurement errors. We know that they exist and are important for the Seine at Paris. Note also that even if the magnitude of these variations is uncertain, the signals are however in phase, which shows that a large part of these variations can still be reproduced by our reconstruction. Note that we now better discuss the limits and uncertainties of the reconstruction in the conclusion.

**p.8 l.4 Which type of correlation is used?**

The Pearson correlation coefficient is used here. We added the precision in the revision of the manuscript.

**Multidecadal hydroclimate variations**
**p.9 – Isn't it difficult to drive conclusions about other variables than P, T or Q as they are difficult to model in hydrological models?**

We use a state-of-the-art physically-based hydrological model with a detailed representation of water exchanges in the soil and resolution of water exchanges at the surface, therefore we have a reasonable confidence in the representation of evapotranspiration and soil moisture. The evaluation shows that we can have a good confidence in the representation of precipitation and river flows. From a surface water budget perspective, it suggests that the other variables of the surface water budget are correctly represented, although it is still possible that some error compensations exist, for example between evapotranspiration and variations in soil moisture.

**Conclusion**
**Wouldn't it be more logical to reverse the conclusion part (for now, first) and the discussion part (second) in the conclusion?**

Agreed. The discussion part on the limit and uncertainties has been improved and moved before the results in the new version of the manuscript. Some perspectives are discussed at the end of the conclusion.

**Orthographic corrections**
**P.1 l.4 – Reformulate the sentence "This method improves the representation of daily flow as well as at longer time step"**
**P.1 l.4 – Provides**
**P.1 l.8 – Maybe "regulate" instead of "modulate" would be better?**
**p.1 l.12 – to influence the drought intensities**
**p.1 l.16 – Missing a "," after "for example"**
**p.1 l.22, p.2 l.11 – Same remark for modulate / regulate**
**p.2 l.1 – Verify the expression "internal variability in climate and/or…"**
**p.2 l.20 – "It" not necessary in "which makes …"**
**p.2 l.25 – "." In the middle of the sentence, between "downscaling" and "of"**
**p.6 l.2 – Maybe add "spatial" before "error". The sentence is not really clear, maybe there is a way to rephrase it, talking about spatial errors for both precipitation and temperature.**
**p.12 l.27 – The SSTs there are**

Thank you, modifications made.

---

## Author Comment (AC2) · 29 Oct 2019

**Response to reviewer#2 of: "Influence of the multidecadal hydroclimate variations on hydrological extremes: the case of the Seine basin" by R. Bonnet, J. Boé and F. Habets.**

First, we would like to thank the second reviewer for his interest in our study and for his general comments, which helped us to improve the structure and the clarity of the manuscript. Major modifications were made in the new version of the manuscript, which is much clearer now. Please find below the point-by-point answers to the comments. For clarity, all reviewer comments are in **bold**.

**Summary:**
**This paper details research conducted on the link between multidecadal hydroclimate variations and streamflow in the Seine basin in France. Unfortunately, I have found it difficult to determine the scientific contribution of this paper due to its flaws in language, layout and lack of detail in the methods. I am recommending that this paper be significantly revised, in order that we may better understand the outcomes of this research. The major points that I recommend are:**

We significantly improved the abstract, the introduction and the conclusion in the new version of the manuscript in order to better highlight the scientific questions and contributions of this work. Important modifications were also made to the method section, which is now more detailed and is much more clearer. An in-depth proofreading was also carried out. These points are discussed with the comments below.

**1. The paper's motivation needs to be better set out. The abstract states that precipitation and groundwater modulate river flows, and that extreme events are influenced by hydroclimate variations. This is not news to the hydrological community. What is novel here? Why is this interesting, and to who, and why?**

We modified the abstract in order to better highlight the motivations and the insights of this work. What is really new in this work is the focus on multi-decadal variations, which are poorly characterized and poorly understood currently.
Please find below the new version of the abstract:

"The multidecadal hydroclimate variations of the Seine basin since the 1850s are investigated. Given the scarcity of long-term hydrological observations, a hydrometeorological reconstruction is developed based on a method that combines the results of a downscaled long-term atmospheric reanalysis and local observations of precipitation and temperature. This method improves previous attempts and provides a realistic representation of daily and monthly river flows. This new hydrometeorological reconstruction, available over a period longer than 150 years while maintaining fine spatial and temporal resolutions, provides an interesting tool to improve our understanding of the multidecadal hydrological variability in the Seine basin, as well as its influence on high and low flows. This long term reconstitution allows analyzing the strong multidecadal variations of the Seine river flows. The main hydrological mechanisms at the origin of these variations are highlighted. Spring precipitation plays a central role by directly influencing the multidecadal variability in spring flows, but also soil moisture and groundwater recharge, which then modulate summer river flows. These multidecadal hydroclimate variations in the Seine basin are driven by anomalies in large scale atmospheric circulation, which themselves appear to be influenced by sea surface temperature anomalies over of the North Atlantic and the North Pacific. The multidecadal hydroclimate variations seem also to influence high flows and low flows over the last 150 years. The analysis of two particularly severe historical droughts, the 1921 and the 1949 events, illustrates how long-term hydroclimate variations may impact short-term drought events, with in particular an important role of groundwater-river

exchanges. The multidecadal hydroclimate variations described in this study, probably of internal origin, could play an important role in the evolution of water resources in the Seine basin in the coming decades. The way in which the associated uncertainties are correctly accounted for in future projections remains to be addressed."

**2. You are missing significant volumes of literature surrounding connections between atmospheric circulation patterns and streamflow across wider Europe. e.g. (among many others)**
**• Folland, C. K., Hannaford, J., Bloomfield, J. P., Kendon, M., Svensson, C., Marchant, B. P., Prior, J., and Wallace, E.: Multi-annual droughts in the English Lowlands: a review of their characteristics and climate drivers in the winter half-year, Hydrol. Earth Syst. Sci., 19, 2353-2375,https://doi.org/10.5194/hess-19-2353-2015, 2015.**
**• Eva Steirou, Lars Gerlitz, Heiko Apel, Bruno Merz, Links between large-scale circulation patterns and streamflow in Central Europe: A review https://doi.org/10.1016/j.jhydrol.2017.04.003**

We acknowledge that a vast body of work exists regarding the connections between atmospheric circulation and streamflows across Europe at the interannual time scales, but it is not the real subject of the paper. Our paper is specifically focused on multidecadal variations, and the problematics are different. We have also largely improved the presentation of the current literature, with a specific focus on multidecadal time scales.

**3. You need to explain your method much more clearly. Can you summarize the method in Bonnet et al (2017) as it is mentioned over and over, and is seemingly critical to the understanding of your method here.**

The method used to develop the hydrometeorological reconstruction is now presented in a dedicated section. The description was entirely rewritten in order to improve its clarity. The method used in Bonnet et al., (2017) is now explained in more details and we improved the description of the method used in this study. We also added a diagram in order to make the understanding of the method easier.

[revised manuscript text omitted]

**4. The paper's language be reviewed by a confident English speaker. The sentence structures are often incorrect, and the tenses are jumbled, which has made the paper very difficult to read. The English appears to improve later in the paper, so I think a little more effort is required.**

We apologize for these language errors; a more in-depth proofreading was carried out in the new version of the manuscript.

**5. The papers structure and headings need to be amended, especially in the methods section.**

We modified the structure in the revised manuscript. As suggested by the third reviewer, we switched the 5th and the 6th section. The subsection about the development of the Seine hydrometeorological reconstruction, originally in the "Data, models and methods" section, was moved in a separate section. These changes improve the clarity of the paper.

**6. Most of your figures have no legends, and many do not have appropriately descriptive axis labels. Please correct this to aid interpretation**

We added legends into the figures and more descriptive axis labels in the new version of the manuscript.

**7. Nearly all of your figures rely on red/green differentiation. 1 in 10 of the male readers will not be able to see this due to common colour-blindness. Please amend your color schemes to avoid this issue.**

Done.

**I have felt unable to go into the detail of the research for these reasons, and would be happy to re-review the paper once these issues have been addressed.**

---

## Author Comment (AC3) · 29 Oct 2019

Response to reviewer#3 of: "Influence of the multidecadal hydroclimate variations on hydrological extremes: the case of the Seine basin" by R. Bonnet, J. Boé and F. Habets.

First, we would like to thank the reviewer for his carefully reading and his interest in our study. The comments greatly helped us to improve the manuscript. Some points are shared with the other reviewers, especially regarding the lack of clarity of the description of the reconstruction method. We made major modifications to the description of the method, which is now in a dedicated section. A diagram was added to help the readers to follow the main steps of the method. Please find below our point-by-point answer to the comments. For clarity, all reviewer comments are in **bold**.

**Summary:**
**Overall this is an interesting paper that develops reconstructed flows for the Seine basin back to the 1850s. Variability in the reconstructions is then assessed and linked to SSTs. While this work will be of interest to the readers of the journal I do not think it is ready for publication. It is difficult to read at present and clarification is needed at times as to what is done. A fuller discussion of the limitations, uncertainties and assumptions of the work is needed. Discussion of other modes of climate variability that could be influential would also be welcome. I highlight some issues below which the authors should address. I hope that these are seen as constructive, as I believe this could be a nice addition to the literature.**

We added a paragraph in the conclusion to talk about the limitations, uncertainties and the assumptions made in this work. We also added some discussions about other modes of climate variability in the introduction, but also we better justify why we are more interested in the Atlantic multidecadal variability (AMV). These points are discussed in more details with the comments below.

**In addition hydro appears twice in the title, while the paper really only deals with SSTs as a source of variability. Suggest title change to "Influence of multi-decadal variability on hydrological extremes: the case of the Seine basin. You could include SST before variability if you wish.**

Thank you for the suggestion. We changed the title to: **"**Influence of multi-decadal variability on high and low flows: the case of the Seine basin**".** We use "high and low flows" instead of hydrological extremes because as you have noted below, we do not directly study the influence of multidecadal variability on floods or droughts, but rather on high and low flows of the Seine basin, even if two droughts are analyzed in details.

**Overall the manuscript needs to be thoroughly edited. There are lots of problems with the use of plurals throughout eg. Precipitations , which should always be singular.**

We apologize for these language errors; a much more in-depth proofreading was carried out.

**It is not clear to the reader what you mean by multi-decadal phases. This appears throughout the paper and I strongly suggest you define use of it early on or use a different term. I assume it means wetter than average or drier than average periods.**

Indeed, a multidecadal phase mean a wetter than average or a drier than average period of at least 20 years. Based on river flows, the positive multi-decadal phases correspond to a succession of years with on average higher flows and conversely. We added this precision in the article, please find below the correction:

"Multidecadal river flows variations at Paris and at Aisy-sur-Armançon are generally in phase. A strong positive multidecadal phase, which corresponds to a succession of years with higher than average river flows, is visible around 1920. On the contrary, a negative phase is present around 1890 and 1960."

"To better understand the mechanisms at the origin of the multidecadal hydroclimate variations on the Seine basin, a composite analysis between the negative multidecadal phases, which correspond to the drier than average periods identified on the Seine river flows at Poses after low-frequency filtering, and positive multidecadal phases, which correspond to the wetter than average periods, is conducted for the main hydrological variables (Figure 7)."

**The abstract states the obvious a little too much, eg. Wet periods are conducive to more flooding, dry periods to droughts. I would be more interested to learn what specifically your reconstructions offer and the new insights they provide. In addition, the final sentence of the abstract gives the impression that this paper looks at dynamical chains in how ssts influence atmospheric circulation. It does not.**

The abstract has been rewritten to better highlight the usefulness of the hydrometeorological reconstruction and the new insights provided by it. Please, find the new abstract below:

"The multidecadal hydroclimate variations of the Seine basin since the 1850s are investigated. Given the scarcity of long term hydrological observations, a hydrometeorological reconstruction is developed based on hydrological modelling and a method that combines the results of a downscaled long-term atmospheric reanalysis and local observations of precipitation and temperature. This method improves previous attempts and provides a realistic representation of daily and monthly river flows. This new hydrometeorological reconstruction, available over more than 150 years while maintaining fine spatial and temporal resolutions, provides an interesting tool to improve our understanding of the multidecadal hydrological variability in the Seine basin, as well as its influence on high and low flows. This long term reconstitution allows analysing the strong multidecadal variations of the Seine river flows. The main hydrological mechanisms at the origin of these variations are highlighted. Spring precipitation plays a central role by directly influencing the multidecadal variability in spring flows, but also soil moisture and groundwater recharge, which then regulate summer river flows. These multidecadal hydroclimate variations in the Seine basin are driven by anomalies in large scale atmospheric circulation, which themselves appear to be influenced by sea surface temperature anomalies over of the North Atlantic and the North Pacific. The multidecadal hydroclimate variations seem also to influence high flows and low flows over the last 150 years. The analysis of two particularly severe historical droughts, the 1921 and the 1949 events, illustrates how long-term hydroclimate variations may impact short-term drought events, with in particular an important role of groundwater-river exchanges. The multidecadal hydroclimate variations described in this study, probably of internal origin, could play an important role in the evolution of water resources in the Seine basin in the coming decades. The way in which the associated uncertainties are accounted for in future projections remains to be addressed"

**The literature review mentions the AMV as an important source of variability, but what about others modes of variability eg. The NAO, East Atlantic Pattern etc. There is more than just the AMV at play and it is my view that this should be reflected in the literature review. These other modes of variability could also be examined for their relationships with the extremes in the reconstructed flows or discounted if the literature has previously addressed these issues.**

We agree that there are other modes of variability influence river flows and hydrological extremes over France, but not necessarily at multidecadal time-scales, which are the center subject of the paper. As our paper focuses on multi-decadal hydrological variations, we limited the literature review to modes of variability important at multidecadal time scales: basically, over France, to the AMV. Indeed, the NAO and EAP are first and foremost interannual modes of variability. Even if Folland et al. (2015) suggest a link between the NAO and the AMV, this link does not seem totally clear (Guan & Nigam 2009). Following your suggestions and the ones of reviewer 2, we added a quick discussion about other modes of variability and we now make it clearer that we are interested in multidecadal variations. We also largely improved the discussion about the link between the AMV and hydrometeorological variations over Europe.

"A large body of work has dealt with river flows variability over Europe. At interannual time scales, the large scale atmospheric circulation over the North-Atlantic plays a major role in hydrological variations over Europe (Kingston et al., 2006b, a; Bouwer et al., 2008; Steirou et al., 2017). The North Atlantic Oscillation (NAO) (Cassou et al., 2004; Hurrell and Deser, 2009) in particular is known to influence river flows over Europe, mainly in winter (Kingston et al., 2006b, a; Bouwer et al., 2008; Steirou et al., 2017).

Regarding the uncertainties in impact projections due to internal variability, variations at longer time-scales, i.e. at decadal and multidecadal time-scales, are far more important than interannual variations, as they may modulate the hydroclimate state on several decades, i.e. at climate time scales. They may temporarily reinforce or reduce, or even reverse, especially over the coming decades, the long-term impacts of climate change."

**In your critique of 20CR on page 2 line 30 you explicitly state that that "this approach is far from optimal, as it does not make use of the long-term meteorological observations that may exist". I would strongly recommend a citation for this or remove.**

The "it" of "it does make use" refers to the reconstruction methods only based on 20CR and does not refer to 20CR. We were talking about local meteorological observations (e.g. precipitation or temperature) that can be used, in addition to 20CR, to obtain the local meteorological forcing necessary for hydrological modelling. Please, find below the new version of the paragraph:

"To move forward, long-term hydrometeorological reconstructions based on hydrological modelling have been developed (e.g Kuentz et al. 2015; Caillouet et al. 2016). Due to the scarcity of meteorological observations in the early 20th century (Minvielle et al., 2015), the meteorological forcing needed for hydrological modelling must first be reconstructed. The recent release of long-term global atmospheric reanalyses (e.g. Twentieth Century Reanalysis (20CR, Compo et al. 2011) from the National Oceanic and Atmospheric Administration (NOAA)) opens great opportunities in that context. Statistical downscaling methods, typically used in climate change impact studies, can be applied to derive the high resolution meteorological forcing necessary for hydrological modelling from these global atmospheric reanalyses, as in Caillouet et al. (2016). This approach presents two main limitations. First, the quality of the reconstruction depends on the quality of the reanalyses. As the density of assimilated observations (e.g. surface pressure in NOAA 20CR, Compo et al. 2011) strongly evolves over time, potential unrealistic trends and/or low frequency variations may exist (Krueger et al., 2013; Oliver, 2016; Bonnet et al., 2017). Second, this approach does not take advantage of the long-term local meteorological observations that may exist."

**There are uncertainties not considered in this paper related to input data, hydrological modelling, other datasets used etc. It would be welcome if the discussion could, in a paragraph, flesh these out.**

Indeed, there is a chain of uncertainty associated with the development of our hydrological reconstruction. We added a long discussion on the uncertainties and limits of the method in the conclusion:

"Although the reconstruction developed in this study is an interesting tool for studying the past variability of the hydrological cycle over the Seine basin, it is obviously not perfect. Uncertainties are present throughout the modelling chain. The statistical downscaling method used at the first step of the reconstruction method assumes that the learning period, over which the large-scale reanalysis and the Safran analysis overlap (1959-2010), is representative of the meteorological conditions of the 1851-2010 period. The consistent performances of the reconstruction over the entire period shown by several analyses in this study suggest that this hypothesis has no major impact on our results. Important uncertainties are associated with the 20CRv2c reanalysis at the beginning of the period, due to the smaller number of assimilated observations (Krueger et al., 2013). We use monthly homogenized local precipitation and temperature observations to constrain the results of statistical downscaling in order to improve the temporal homogeneity of the reconstruction, but the homogenization method is not state-of-the-art. The good agreement between the low-frequency variations of the homogenized monthly precipitation series and of the Global
Precipitation Climatology Centre dataset (Schneider et al., 2008) from 1901 to 2011 (not shown) still gives good confidence in the overall realism of the multidecadal variations described in this study.
The hydrological model used in this work is also not perfect, with potential consequences on our results. The amplitude of the variations in the piezometric levels at Toury in particular is strongly underestimated. Although this is a one-point measurement, it is possible that this underestimation exists for the entire Beauce aquifer, which could imply an underestimation of the multidecadal variability of reconstructed river flows, especially in summer when the role of groundwater is particularly important. This underestimation is likely due to hydrological modelling, and could come from a too simple representation of the limestone aquifer that neglects some confined parts. The river channel dimensions and conveyance capacity are also supposed constant over time, whereas they could be affected by multidecadal climate variability (Slater et al., 2019).
Some important uncertainties are also associated with the observations used for the evaluation of the reconstruction. They may indeed be influenced by non-climatic anthropogenic influences such as dams and pumping, not taken into account in the hydrological model, or changes in measurement methods for example. We have evaluated the reconstruction against multiple observations in an effort to ensure that our results are not too dependent on a particular source of information and to better understand the potential limitations of the reconstruction."

**As mentioned, the methodology is complex and difficult to follow. The use of analogues to derive daily precipitation seems to move from 2800 samples to 60 to 3. Why three, why such a decrease in sample size and are you left with a large enough sample?**

We totally rewrote a large part of the method section in the revised manuscript, by adding some explanations, in particular on the method used in Bonnet et al. (2017) (as asked below in a comment consistently with the other reviewers), on which we based the new method developed here. It should be clearer now. See also our answer to following comments below.

To respond to your comment, an important objective of our hydrometeorological reconstruction is to improve the representation of daily river flows. As a result, a balance has to be found for the number of analogue days to be used during the monthly constraint. The greater the number of analogue days used for the monthly constraint is, the farther some analogues are from the target day, with likely in the end a degradation of the representation of precipitation and temperature, and therefore of river flows. On the other hand, too few analogue days could limit the improvement in low frequency variations expected from the monthly constraint. We therefore made different tests in order to find the best number of analogues to retain at the different steps.

[Figure]

*Figure R1: Spatial distribution of the correlations between (a-c) daily and (b-d) monthly (a-b) precipitation and (c-d) temperature over the Seine basin. Correlations are calculated between the Safran analysis (considered as observations) and the precipitation and temperature derived from (brown) the reconstruction developed in Bonnet et al., 2017, (green) the downscaling method alone, (purple) the downscaling method only constrained by monthly precipitation and temperature, (blue) the downscaling method only constrained by daily precipitation and (red) the downscaling method constraint by daily and monthly precipitation and temperature, based on different tests for the number of analogs used for the monthly constraint (X axis). The correlations are calculated on the 1958-2005 period and the series have been deseasonalyzed beforehand. The boxplots show the minimum/25th percentile/median/75th percentile and the maximum.*

This is illustrated in figure R1, which shows the results of one of these tests. It shows that the daily correlations between reconstructed and observed temperature and precipitation decrease when the number of analogue days increase (Figure 1 a and c). After testing different possibilities, we decided to keep the 3 best analogue days from the daily constraint. These 3 analogue days are then used to apply the monthly constraint. With 3 analogues, the ensemble is large enough for the monthly constraint.

Figure R1 also shows that the double constraint method, at daily and then monthly time scales, greatly improves the daily correlations of precipitation and temperature compared to the statistical downscaling method alone, or to the downscaling method only constrained at monthly time scale (Figure 4.11 a and c).

**Similarly there are assumptions made that things like channel dimensions and conveyance capacity remain stationary over time. Recent research has shown that they also react to variability Slater, L.J., Khouakhi, A. and Wilby, R.L., 2019. River channel conveyance capacity adjusts to modes of climate variability. Scientific reports, 9(1),pp.1-10.**

Thank you for this interesting publication. Indeed, our hydrological model, as most hydrological model used in the literature, doesn't take into account the possible changes in channel dimensions or conveyance capacity due to multi-decadal climate variations. We added this information in the paragraph that now discusses the different uncertainties (see two points above).

**In terms of your methods does your use of analogs constrain the variability to that of the shorter record? The methodology is complex and hard to follow as it is written. Previously published work on which this data is based needs to be shortly summarised. In addition it would be helpful to the reader to provide a flow diagram of the key steps in the workflow. For example, I have read the paper multiple times and at the start of the results I am unsure what the reference simulation is.**

We acknowledge that our method section was not clear enough and needed to be improved. Indeed, the final variability is limited by that of the shorter record, as in the end, an analogue is used. We added a paragraph that summarizes the main ideas of our method in order to clarify the method section (below). We also added a diagram to clarify the explanation of the method. Please find the new method section below.

[revised manuscript text omitted]

**I don't like the word anthropized – heavily impacted would work just as well.**

We changed the word anthropized to "heavily impacted by human activities".

**It appears that the reconstructions persistently underestimate the annual maxima (fig 4). What does this say about your reconstruction, does it affect your results and how might this be improved?**

The annual maxima are very similar in the reconstruction and in the reference simulation based on the same Safran-Surfex-AquiFR hydro-meteorological system over their common period (1959-2014). Therefore, the differences of the annual maxima are not related to the quality of the reconstructed meteorological forcing. They may be due to the hydrological model, which would cause a general underestimation of the annual maxima (it is true the annual maxima are closer to the observations after 1960, but it is likely the result of the construction of flood-controlling dams, not taken into account by the hydrological model).

A mean bias does not affect much our results as we are mostly interested by the variability and the variability of the annual maxima is well reproduced, over the entire period. Note also that the selection of high flows is based of percentiles rather than in terms of absolute river flows. The historical flood of 1910 for example is the biggest event in our reconstruction and in the observations.

**Have the piezometric level data been quality assured?**

A fairly high level of confidence is placed in the piezometric observations of the Beauce groundwater at Toury. Indeed, the piezometric station at Toury is managed by a sugar beet factory consistently over decades. As one of the oldest dataset of piezometric head over France, it is used as a reference by water management agencies, especially the one that manage the beauce aquifer. It is, therefore, validated by experts. Additionally, the observations at Toury in the recent period are consistent with the ones observed at nearby stations, which give us also confidence on this series. If some uncertainties on the change of measurement method could exist, it hasn't a strong influence on the observations quality as the several meters variations observed are far above the order of magnitude of traditional measurement error.

**It is stated on page 8 line 29 that "The length of the reconstruction allows to show that multidecadal variations are also present before the 20th century....." Is this not expected. In the same sentence what do you mean by negative phase.**

By negative phase, we mean a succession of years with lower than average river flows. The fact that the reconstruction shows multi-decadal variations before the 1900s was not obvious and is an interesting result. The long-term available river flow observations show the beginning of a negative multidecadal phase in this period (Figure 3c-d), but it is difficult to affirm something regarding this period based only on observations, due to observational uncertainties. Moreover, the internal climate multidecadal variability (that is assumed here as the main driver of these variations) in the North Atlantic region might not to be constant over time (Qasmi et al., 2017). The context, which justifies the interest of this result, was not well described in the first version of the manuscript. We improved that point in the new version.

"Annual river flows of the Seine at Poses (Figure 1) from the reconstruction show strong multidecadal variations during the 20th century (Figure 7), consistently with the observed variations over France described in Boé et Habets (2014). As said in the introduction, major sampling uncertainties exist when dealing with multidecadal variations on the short observational record. Additionally, since the mid-20th century, the climate has been strongly

impacted by anthropogenic forcings, making it difficult to disentangle the respective role of internal variability and external forcings in observed hydroclimate variations.

It is therefore very interesting to note, thanks to the extended length of our reconstruction compared to previous works, that multidecadal variations also exist before the 20th century, with lower than average river flows around the 1885-1905 period (about 15\% lower than the average of 446m3/s for annual river flows). It reinforces our confidence in the reality of multidecadal hydrological variations in France observed over the 20th century and in the idea that they are at least partly of internal origin. Phased multidecadal variations also exist for the seasonal averages, with the strongest absolute variations seen in spring and winter. Note however that as climatological river flows are smaller in summer and early fall, the multidecadal variations seen in these seasons are also important."

**Throughout you could quantify these above and below average phases in terms of the magnitude of their anomaly.**

A quantification of the intensity of these phases was carried out in section 4.1 of the manuscript, where the relative difference between the negative and positive multidecadal phases for the four seasons are calculated. Variations ranging from 30 to 40% are visible in summer, for example, which can have serious impacts in terms of water resources. Therefore, the positive and negative multidecadal phases broadly correspond to variations of 15 to 20% in relation to the anomaly of the Seine river flows.

**Indeed what can your work tell us about an appropriate baseline for assessing changes in the Seine flows – one that accounts for the types of variability you see. Thirty year baselines often only sample one component of multi-decadal variations.**

This work indicates that considering a baseline of only 30 years, to dimension structures such as drinkable water plants or irrigation dams, is clearly insufficient and even dangerous from a practical perspective. The use of periods of at least 60 years would be much better. We added this interesting point on the new version of the conclusion.

"The results described in this paper illustrate how dangerous it can be for practical purposes to rely on short periods of few decades (e.g. three) to characterize hydrological hazards."

**There are a number of statements that require clarification or a more precise wording, such as: multi-decadal phases? P7 line 10 over what period are correlations derived? Page 8 line 23 A partially captive part of the aquifer is not represented in the model – what is a partially captive part? Pg 11 line 11 a large number of stations have a ratio between one and two. . ..with a ratio of 1.1 consistent with the reconstruction in that region. I can't interpret this. Pg 11, line 24 strong positive multidecadal phase?**

Clarifications have been made in the text on these different points. For the first point, which is probably the most important one, a more precise definition of "multidecadal phases" is now given (see also a previous comment).

For the second point, the period considered for the correlation is indicated in the legend of the figure, but we added it in the text to be clearer:

"The medians of the correlations with observations over the 1958-2005 period are respectively of 0.7 and 0.97 for daily and monthly river flows (Figure 3)"

Regarding the third point, a part of the Beauce aquifer contains lenses of clay, which divide the aquifer in a confined and unconfined part, and modify the hydrodynamic of the aquifer flow. These lenses of clay are not represented yet within the model because of his too coarse spatial resolution. This too simple representation of the limestone aquifer that neglects these confined parts could induce the underestimation of the piezometric levels variations. We added this clarification in the article.

"A partially captive part of the limestone aquifer, which contains lenses of clay, is not represented in the model due to his insufficient resolution. This part of the aquifer amplifies the flow time and, therefore, the memory of the aquifer."

For the fourth point, we changed the second part of the text, which could be confusing:

p11, l24: "The ratio of low flow days between negative and positive multidecadal phases is indeed between one and two for a large number of stations (Figure 13b). For the observations at Aisy-sur-Armançon (triangle located in the south east of the basin) the ratio is 1.1, consistent with the reconstruction."

For the last point, as for the first point, the strong positive multidecadal phase mean a wetter than average period. We choose here two major historical droughts, which occur during opposite phase of the multidecadal variability of the Seine river flows. This is interesting to look at the impact of the multidecadal variability describe before on particular extreme events.

**Is it fair to call a day with flow above the 95th percentile a flood day. Is it fairer to say a high flow day? The same issue applies to drought days, are these more fairly described as low flow days? In reality you are not strictly looking at floods and droughts in this part of the analysis.**

It is true that the flows above the 95th of below the 5th percentile are not rare enough to be qualified as flood or drought days. We changed these terms "floods" and "droughts" to high and low flows in the revised version of the manuscript.

**It would be useful to outline a potential or plausible chain of causation as to how SSTs in the North pacific relate to Seine flows.**

The potential mechanisms linking SST anomalies are still not well understood. We added some information on that point in the new version of the manuscript (below).

"Significant SST anomalies between the negative and positive multidecadal phases of the Seine river flows are also observed in the North Pacific. SSTs there are significantly warmer during the negative multidecadal phases of Seine river flows compared to the positive phases. The potential mechanisms linking the SST anomalies in the North Pacific and multidecadal hydroclimate variations over France are not clear. Ding et al. (2017) suggest that the Pacific decadal variability (PDV) (Mantua and Hare, 2002) combined with La Niña events could result in precipitation and temperature anomalies over Europe by favoring NAO-like circulation patterns. The phase of the NAO and therefore the sign of precipitation and temperature anomalies is dependent on the type of La Niña. Note also that the apparent link between the North Pacific SSTs and Seine river flows might not be causal. Indeed, the SST anomalies in the North Pacific might be to a certain extent also driven by the AMV. Ruprich-Robert et al. (2017) indeed suggest that the AMV may influence North Pacific SSTs, mainly through an atmospheric teleconnection originating from the tropical Atlantic."

**Do your extended flows add additional insight into AMV related variations in flows than done in previous work. Has the magnitude of anomalous periods been the same or different in your reconstructions relative to previous work?**

Indeed, our hydrometeorological reconstruction allows adding additional insights to our comprehension of the link between the AMV and the river flows. The first contribution of this work is that it allows to confirm on a longer period the hypotheses made by Sutton and Dong (2012) and Boé and Habets (2014). By using a longer period of study, we are able to use two more multidecadal phases in our analysis, which makes it more robust. Our work also highlights that this influence is not limited to river flows over France, but also concerns high and low flows over France.

Note that a new analysis has been added to section 6 on the "Role of large-scale circulation and influence of ocean variability", with interesting new results.

Please find below the new analysis and the figure associated:

[revised manuscript text omitted]

**Section 4.1 is hard to follow, it might be worthwhile organising by season. At times the text jumps from season to season**

Agreed. Modification made.

**I think section 6 needs to come earlier in the paper, given it importance to the message of the paper.**

Agreed. The section 6 and the section 5 have now been switched.

---

## Referee Report (RR1)

The authors bring answers to the previous comments made by the three reviewers. The pertinence of the paper is in the scope of the journal and results remain interesting. The addition of a discussion section, as well as the detail of the methodological framework are appreciated. There are still some points to clear up, but mainly from the form and not the content. There are still some sentences that are grammatically incorrect and improving the language would improve the fluidity as well as the understanding.

My main comments are about the form of the methodological framework (need subsections and reorganization), as well as the lack of description of the figures (which can be hard to understand, if we have only the legend and the interpretation). I also have a question about the relevance of transforming a probabilistic analogue method to a deterministic one.

**General comments:**

**Methodological framework – Section 3:**

**1)** I really had a hard time understanding the overall process of the method. I think it would be more understandable with subsections. As I understood:

p.6 l.13-18 = Sort of introduction

p.6 l.19 to p.7 l.2 = The analogue method (generalities) with a part of the setup (analogy domains)

p.7 l.3 to p.7 l.10 = How daily time series are created from the analogue method (outputs)

p.7 l.11 to p.7 l.19 = The monthly constraint. Here you mention the Step 3 of the method, but Steps 1 and 2 have not been mentioned before, which is confusing

p.7 l.20 to p.7 l.27 = Advantages of this method

p.7 l.28 to p.7 l.31 = Here, we begin with the setup, so the global reanalysis used (20CR). Isn't it possible to put that with the analogue method? Or to put all the setup together in the same subsection?

p.7 l.32 to p.8 l.2 = Step 2 (after Step 3 in the text)

p.8 l.3 to p.8 l.20 = Daily constraint, so Step 2. Why isn't it just after "how daily time series are created"?

p.8 l.21 to p.8 l.26 = We are talking about the monthly constraint again (step 3), it definitely lost me.

p.8 l.27 to the end of Section 3 = Conclusions

As you can see, I really had to separate the different paragraph to understand the chronology, which is not in coherence with the one if your Figure 2. Maybe you can regroup in subsections by Step (Intro / Step 1 / Step 2 / Step 3 / Conclusion) or add a subsection about the setup (which analogy domain, which reanalysis, etc..). And really use your figure 2 and describe it in the text. It is up to you for the subsections, but in my opinion, the form of this section can easily be improved with reorganization.

**2)** About your method, as I understand, at the end, you have a deterministic reconstruction. Why not keeping a probabilistic dataset to consider the uncertainties associated with the downscaling? Is it really relevant to select the best analogue day based on daily/monthly constraints, and not keep the 3, 5 or 10 best analogues at the end of the last step? Maybe you would have very different signals in your reconstruction, just with the 2$^{nd}$ best analogue?

**3)** Interpretation of figures is well discussed. However, there is a lack of precision to really understand the figures and what they represent. This is why it took me a long time to really appreciate your results. It is necessary to look at the legend, as well as the text, several times to catch up the idea behind the figure. Would it be possible, for each figure, to really explain the computations behind, with what data / what score, etc., before interpreting it? Like "Figure X represents …" (computation), "It shows…" (factual description of the figure), "It means…" (interpretation)? The units are also missing in a lot of figures (% in Figure 8 for example) and legends are not totally clear.

**Specific comments**:

P1 l.19: "The" in "The future impacts of climate change" could be removed

p.1 l.21: Is the strong reduction of river flow in summer detected everywhere?

p.1 l.23-24: "…and regarding precipitation". The sentence seems not finished. Did you simply mean "for precipitation"?

p.2 l.19: "The river flows variations" -> the river flow variation, be careful with these plurals (several times in the text, same p.4, l.16)

p.2 l.24-25: "which he attributes to change in atmospheric 25 circulation likely associated with the AMV" -> Please, check the grammar

p.3 l.13: "which makes very difficult" -> remove the "it"

p.3 l.34: Not sure that "varied" is the good word here. What do you mean? Observations of different variables contributing to the hydrological cycle?

p.3 l.35: "of high and low flows" -> remove the "the"

p.4 l.1: Not sure that "exhibit" is the good word here.

p.8 l.28 29: Prefer "These tests concern, for example, the best combination of weights given to precipitation and temperature errors **or** the number of analogs selected at each step (without the s)." Or develop the "etc."

p.9 l.6: "Simply" could mean a lot of things! How do they are corrected?

p.9 l.13: Remove the "s" to "the medians", same problem with the plural forms.

p.9 l.13: The median across what? Different observations? You have one series of observation and of reconstructed. So, what is about the median? You don't really explain how this figure is constructed, and I am not sure I understood it.

p.9 l.30: "The meteorological forcing of the reconstruction is very likely not responsible for the potentially unrealistic trend in river flows, as reconstructed precipitation and temperature do not show unrealistic trends (not shown)." This sentence means a lot, and I am not sure to well understand it. Can you rephrase?

p.11 l.4: You're talking about Figure 7 for annual river flows, but it shows low pass filtering, so is it the good figure?

p.13 l.13: You've explained that the 2/3 of the variations come from groundwater, and not precipitation, but maybe is it related?

p.16 l.13: "focusing"

References: Bindoff, Casanueva Vicente and Moisselin: Journal missing?

---

## Referee Report (RR2)

Thank you for your corrections to the paper "Influence of multidecadal variability on high and low flows: the case of the Seine Basin". The paper is much improved and now reads with a lot more clarity than before. Section 3 reads much better, Section 4 is also well written, and the conclusion is good. I suggest the paper be accepted subject to minor revisions:

General comments:

1. I'm not sure enough focus is given to the analysis of the extremes, given that is the main novelty of the approach and paper over Bonnet et al 2017.
2. I'd be tempted to end the abstract on a more positive note
3. You mention that summer and fall flows are lower, but the variations are also important. Perhaps Figure 7 would be better plotted as percentage anomalies to show this?
4. Section 5.2 – how are groundwater-river exchanges quantified?
5. Section 6 – first sentence – "they are mainly linked to variations in precipitation" – I'm not sure this is the message you portray in the previous section. You state that "in summer, unlike in spring, no significant differences in precipitation between positive and negative multidecadal phases are noted" and "in winter… slight differences in precipitation……are not significant" etc. You also conclude that "Large and significant multidecadal variations in summer and 20 winter river flows exist, which cannot be explained by concomitant variations in precipitation". So you should clarify this statement somehow (as you do in the conclusion, page 17 lines 22-23), are you only referring to spring? Otherwise, it doesn't add up!

Figure corrections

1. Figure 1 – could you choose better colouring? the red and magenta are very similar
2. Figure 3 – please relabel the x axes "1", "2", "3" instead with "reference", "20CRpt" and "Seine" for readability. Please also label the Y axis. The dots appear black, not blue, but their colour doesn't matter. The caption is missing an "s" on the penultimate line from "values"
3. Figures 6 and 7 – I would relabel the y axes to say they're anomalies
4. Figures 7 and 9 – this is not a colour-blind friendly palette – the Fall and Spring are not differentiable. Making the green paler and the red brighter and darker would help (see image below)
5. Figure 8 – please label the columns and rows on the graphic itself with the variables along the top and the seasons along the right hand side to aid interpretation. I would also be tempted to reorder the rows to read spring at the top, then summer, autumn, and winter, in the order you discuss them
6. Figure 12 – I don't think I understand this graph and its implications. The caption is VERY long. Should some of this description be in the body of the paper?
7. Figure 13 – Is it sensible to reverse the scale from figs a and b? maybe use colours that don't signify "wet" and "dry"
8. Figure 14 – caption should read mm/d not mm/j
9. Figure 14 – please add the crosses to the legend as 1910-1930 and 1940-1960.

C

[Figure]

ass filtered interannual anomalies of the Seine river flows at the Poses station. Black: annual, blue: winter, and brown: autumn. The annual mean over the whole period is 446m3/s.

P

ass filtered interannual anomalies of the Seine river flows at the Poses station. Black: annual, blue: winter, and brown: autumn. The annual mean over the whole period is 446m3/s.

10.

Grammar corrections:

- Throughout – should multidecadal be hyphenated: multi-decadal?
- Page 2 line 1 – "correctly reproduce", not "reproduce correctly"
- Page 2 line 4, line 19 – "river flow variability", not "flows"
- Page 2 line 11 – "for several decades", not "on"
- Page 2 line 33 – "for the second half", not "on"
- Page 4 line 1 – "this allows to study" isn't correct, I suggest "this allows studies into whether"

- Page 4 line 3 – "this study therefore has" not "has therefore"
- Page 4 line 16, Page 9 line 15 – "flow" not "flows"
- Page 5 line 1, line 9 – "platform" not "plateform"
- Page 5 line 2 – what does ISBA stand for?
- Page 5 line 12 – "based on thousands of observation stations"
- Page 5 line 32 – im not familiar with what "weights" means
- Page 6 line 26 – "searched for"
- Page 7 – make sure your use of commas in numbers in consistent. There is 5000 and 5,000 on the same page. Find out the HESS standard
- Page 7 line 24 – recommend rewording to "this approach additionally takes local observations into account in the downscaling process…"
- Page 8 line 31 – "degrades"
- Page 8 line 32 – "developed on the Seine basin", suggest changing on to "over" or "for"
- Page 9 line 15 – "even almost as well reproduced" doesn't read well, I suggest something like "river flow variability is reproduced almost as effectively as the reference simulation"
- Page 9 – I don't think you need to keep re-referencing Fig 3 as often.
- Page 9 line 28 – suggest replacing "even if" with "although"
- Page 10 line 25 – "by a factor of 2"
- Page 10 line 27 – I think HESS prefer not to quote personal communication, as you say "might be linked", I think it's fair to leave the reference out of this sentence.
- Page 11 line 18 – do you mean to reference Figure 8?
- Page 13 – please write NAO, AMV and SST in full, the abbreviations were last seen on page 2, a long way away!
- Page 13 line 19 – "circulation anomalies on a few decades" should possibly be "circulation anomalies for a few decades" but I don't fully understand the statement
- Page 14 line 13 – would we call this "paleoclimate"? It seems too modern to be considered paleo but I might be wrong
- Page 17 line 27 – "longer" not "longest"
- Page 18 line 18 – "progress" not "progresses"
- Page 18 line 23 – suggest "climate models may have difficulties capturing some properties"

---

## Author Response (AR2)

**Response to reviewer of: "Influence of multidecadal variability on high and low flows: the case of the Seine basin" by R. Bonnet, J. Boé and F. Habets.**

We want to thank the three reviewers for their careful reading and the many comments and suggestions that helped us to improve the clarity of our paper. We corrected the language errors still present in the text and apologize for that. Please find below the point-by-point answers to the comments. For clarity, the reviewers' comments are in **bold**.

**Response to #reviewer1:**

**Summary: The authors bring answers to the previous comments made by the three reviewers. The pertinence of the paper is in the scope of the journal and results remain interesting. The addition of a discussion section, as well as the detail of the methodological framework are appreciated. There are still some points to clear up, but mainly from the form and not the content. There are still some sentences that are grammatically incorrect and improving the language would improve the fluidity as well as the understanding.**
**My main comments are about the form of the methodological framework (need subsections and reorganization), as well as the lack of description of the figures (which can be hard to understand, if we have only the legend and the interpretation). I also have a question about the relevance of transforming a probabilistic analogue method to a deterministic one.**

We reorganized the section three and added four subsections. The description of some complicated figures has been improved.

**General comments:**
**Methodological framework – Section 3:**
**1) I really had a hard time understanding the overall process of the method. I think it would be more understandable with subsections. As I understood:**
**p.6 l.13-18 = Sort of introduction**
**p.6 l.19 to p.7 l.2 = The analogue method (generalities) with a part of the setup (analogy domains)**
**p.7 l.3 to p.7 l.10 = How daily time series are created from the analogue method (outputs)**
**p.7 l.11 to p.7 l.19 = The monthly constraint. Here you mention the Step 3 of the method, but Steps 1 and 2 have not been mentioned before, which is confusing**
**p.7 l.20 to p.7 l.27 = Advantages of this method**

**p.7 l.28 to p.7 l.31 = Here, we begin with the setup, so the global reanalysis used (20CR). Isn't it possible to put that with the analogue method? Or to put all the setup together in the same subsection?**
**p.7 l.32 to p.8 l.2 = Step 2 (after Step 3 in the text)**
**p.8 l.3 to p.8 l.20 = Daily constraint, so Step 2. Why isn't it just after "how daily time series are created"?**
**p.8 l.21 to p.8 l.26 = We are talking about the monthly constraint again (step 3), it definitely lost me.**
**p.8 l.27 to the end of Section 3 = Conclusions**
**As you can see, I really had to separate the different paragraph to understand the chronology, which is not in coherence with the one if your Figure 2. Maybe you can regroup in subsections by Step (Intro / Step 1 / Step 2 / Step 3 / Conclusion) or add a subsection about the setup (which analogy domain, which reanalysis, etc..). And really use your figure 2 and describe it in the text. It is up to you for the subsections, but in my opinion, the form of this section can easily be improved with reorganization.**

We modified the structure of the methodological framework section by adding subsections, as you suggested. This section is clearer now. Please find the new section below.

[revised manuscript text omitted]

**2) About your method, as I understand, at the end, you have a deterministic reconstruction. Why not keeping a probabilistic dataset to consider the uncertainties associated with the downscaling? Is it really relevant to select the best analogue day based on daily/monthly constraints, and not keep the 3, 5 or 10 best analogues at the end of the last step? Maybe you would have very different signals in your reconstruction, just with the 2nd best analogue?**

There are two main reasons why we decided to develop a deterministic rather than a probabilistic hydrometeorological reconstruction. First, one of our objectives was to reconstruct as well as possible the past hydrological cycle of the Seine basin with the

available data/models, in particular to see whether or not we could find the observed multidecadal variations in our reconstruction, and to study the associated physical mechanisms. By keeping several analogues at the end of the method, the associated trajectories would have been degraded compared to the first one, which is not really of interest in our case. The second reason is a more technical one. We chose to use a physics-based model, which is interesting because it gives us access to variables such as soil moisture or groundwater exchanges among others. However, these types of model are costly in terms of calculation time, and it is thus impossible to carry out many simulations.

**Results:**
**3) Interpretation of figures is well discussed. However, there is a lack of precision to really understand the figures and what they represent. This is why it took me a long time to really appreciate your results. It is necessary to look at the legend, as well as the text, several times to catch up the idea behind the figure. Would it be possible, for each figure, to really explain the computations behind, with what data / what score, etc., before interpreting it? Like "Figure X represents …" (computation), "It shows…" (factual description of the figure), "It means…" (interpretation)? The units are also missing in a lot of figures (% in Figure 8 for example) and legends are not totally clear.**

We apologize for the missing units in some of the figures. We added the units missing and modified some of the figures and their captions for more clarity. We also add some explanations in the text to facilitate the comprehension of some figures. Please find below these explanations and find the modified figures and their captions in the revised version of the manuscript.

Figure 3:
"First, the temporal correlations between observed river flows and river flows from the Bonnet et al. (2017) reconstruction ,the present reconstruction and the reference simulation are calculated at the daily and monthly time scale for the 1958-2005 period at the 136 gauging stations available over the Seine basin (Section 2.1). The median of these temporal correlations is respectively of 0.7 and 0.97 for daily and monthly time scales (Figure 3) The daily and monthly river flow variations are therefore correctly captured in the Seine reconstruction."

Figure 5:
"The annual maximums of daily river flows are computed for the Seine reconstruction, the reference simulation and the observations for the Seine at Paris and compared (Figure 5)."

Figure 9:

"The groundwater-river exchanges from the reconstruction, calculated by the AquiFR hydrogeological model, show strong multidecadal variations over the past 150 years for all seasons (Figure 9)."

Figure 10:
To assess whether soil moisture may play a role in summer river flow variations, the relative changes in the ratio of total runoff to precipitation between the positive and negative multidecadal phases is calculated in summer with the reconstruction (Figure 10).

Figure 11a:
"The role of large scale atmospheric circulation, an important driver of precipitation is first studied. A composite analysis of sea level pressure during the positive and negative multidecadal phases of the Seine river flows previously defined is conducted. As in section 5.1, the significance of changes between negative and positive multi-decadal phases is assessed with a Student's t-test and p-value < 0.05."

**Specific comments:**
**P1 l.19: "The" in "The future impacts of climate change" could be removed**
Done

**p.1 l.21: Is the strong reduction of river flow in summer detected everywhere?**
Yes, the strong reduction of summer river flows is detected everywhere in France. We modified the sentence in the new version of the manuscript:

"In France, important hydrological changes are expected at the end of the 21st century in response to global warming, with for example a strong general reduction of river flows in summer (Dayon et al., 2018)"

**p.1 l.23-24: "...and regarding precipitation". The sentence seems not finished. Did you simply mean "for precipitation"?**
Yes, we changed "regarding precipitation" to "for precipitation"

**p.2 l.19: "The river flows variations" -> the river flow variation, be careful with these plurals (several times in the text, same p.4, l.16)**
Done

**p.2 l.24-25: "which he attributes to change in atmospheric 25 circulation likely associated with the AMV" -> Please, check the grammar**
We apologize for the grammar error, we corrected it in the revised manuscript:

"Willems (2013) notes the existence of multidecadal variations in extreme precipitation and river flows over Europe, including France, attributed to changes in atmospheric circulation likely driven by the AMV"

**p.3 l.13: "which makes very difficult" -> remove the "it"**
Done

**p.3 l.34: Not sure that "varied" is the good word here. What do you mean? Observations of different variables contributing to the hydrological cycle?**
Yes, we mean different variables related to the hydrological cycle. We changed it in the new version of the manuscript:
"The interest of this basin is the existence of long observational series of different variables contributing to the hydrological cycle, which are useful for developing and evaluating the reconstructions."

**p.3 l.35: "of high and low flows" -> remove the "the"**
Done

**p.4 l.1: Not sure that "exhibit" is the good word here.**
We changed "exhibit" by "are influenced by"

**p.8 l.28 29: Prefer "These tests concern, for example, the best combination of weights given to precipitation and temperature errors or the number of analogs selected at each step (without the s)." Or develop the "etc."**
Done

**p.9 l.6: "Simply" could mean a lot of things! How do they are corrected?**
We added precisions on how we corrected the biases in the new version of the manuscript:
"These mean climatological biases are corrected before the hydrological modeling by calculating for each season of the study period the mean biases with Safran. Then, for each seasons (e.g. winter 1852, spring 1852, summer 1852...), the bias is corrected proportionally at the hourly time step."

**p.9 l.13: Remove the "s" to "the medians", same problem with the plural forms.**
Done

**p.9 l.13: The median across what? Different observations? You have one series of observation and of reconstructed. So, what is about the median? You don't really explain how this figure is constructed, and I am not sure I understood it.**
We added some precisions in the revised manuscript to facilitate the comprehension. Please, find the new paragraph in your third point about the results, figure 3.

**p.9 l.30: "The meteorological forcing of the reconstruction is very likely not responsible for the potentially unrealistic trend in river flows, as reconstructed precipitation and temperature do not show unrealistic trends (not shown)." This sentence means a lot, and I am not sure to well understand it. Can you rephrase?**

We rephrased that sentence in the revised manuscript:

"It is very unlikely that a potentially unrealistic trend has been introduced in hydrological reconstruction by the meteorological forcing developed in section 3, as the reconstructed precipitation and temperature series do not show unrealistic trends (not shown)."

**p.11 l.4: You're talking about Figure 7 for annual river flows, but it shows low pass filtering, so is it the good figure?**

Yes, we talk about the Figure 7. We added "Low-pass filtered annual river flows" instead of "annual river flows" in order to clarify this point.

**p.13 l.13: You've explained that the 2/3 of the variations come from groundwater, and not precipitation, but maybe is it related?**

Indeed, the ⅔ of the summer river flows variations come from groundwater, as there are no significant concomitant variations of precipitation. However, these multidecadal variations in groundwater are very likely due to variations in the recharge induced by the multidecadal variations noted previously in spring precipitation. Therefore, the summer river flows variations are indirectly influenced by these multidecadal variations in spring precipitation. We added a precision on this point in the new version of the manuscript:

"As variations in groundwater-river exchanges, very likely induced by spring precipitation, roughly explain 2/3 of the strong multidecadal river flows variations that exist in summer, other mechanisms are involved."

As this point is also explained on the section just before, we did not add more information.

**p.16 l.13: "focusing"**
Done

**References: Bindoff, Casanueva Vicente and Moisselin: Journal missing?**
Done

**Response to #reviewer2:**

**Thank you for your corrections to the paper "Influence of multidecadal variability on high and low flows: the case of the Seine Basin". The paper is much improved and now reads with a lot more clarity than before. Section 3**

**reads much better, Section 4 is also well written, and the conclusion is good. I suggest the paper be accepted subject to minor revisions:**

**General comments:**
**1. I'm not sure enough focus is given to the analysis of the extremes, given that is the main novelty of the approach and paper over Bonnet et al 2017.**
Another important aspect of this paper is to show that this multidecadal variability in flows is not limited to the 20th century, although its intensity and timing may vary, and to confirm the role of the North Atlantic sea surface temperature in these variations, as well as to identify the associated hydrological processes. This finally provides information on how the high and low flows are influenced by the multidecadal variability of the Seine hydrological cycle. In this sense, we believe that we gave the analysis of the high and low flows enough focus in the paper.

**2. I'd be tempted to end the abstract on a more positive note**
The third reviewer also asked for a modification of the end of the abstract. In order to improve it, we changed the last sentence in the revised manuscript:

"It is therefore essential to take the associated uncertainties into account in future projections."

**3. You mention that summer and fall flows are lower, but the variations are also important. Perhaps Figure 7 would be better plotted as percentage anomalies to show this?**
We modified Figure 8, which is now in percentage. As it represents the relative differences between the negative and positive multidecadal phases of the Seine river flows, the importance of the variations in summer and fall are already visible on this figure.

**4. Section 5.2 – how are groundwater-river exchanges quantified?**
The exchanges between groundwater and river are computed daily. They depend on the gradient head between the river and the aquifer, and to a transfer coefficient that accounts for the river bed characteristic. A detailed explanation of the calculation is provided by Vergnes & Habets 2018. To sum up, at each time step and for each river cell, a river level is estimated based on the river flow. These rating curves are first estimated at the gauging station and then extrapolated along the river network. Then, the groundwater-river exchanges are estimated while resolving the diffusivity equation to estimate the groundwater head. The estimation is eased by the fact that the grid of the aquifer domain fit the river network. Examples of the river aquifer exchange are also given in Vergnes et al., 2020.

We added a precision on this point in the subsection 2.2 "The Safran-Surfex-AquiFR hydrometeorological system" of the revised version of the manuscript:
"The exchanges between groundwater and river are computed daily based on the

gradient head between the river and the aquifer, and a transfer coefficient that accounts for the river bed characteristic (see details in Vergnes et Habets 2018)."

**5. Section 6 – first sentence – "they are mainly linked to variations in precipitation" – I'm not sure this is the message you portray in the previous section. You state that "in summer, unlike in spring, no significant differences in precipitation between positive and negative multidecadal phases are noted" and "in winter... slight differences in precipitation......are not significant" etc. You also conclude that "Large and significant multidecadal variations in summer and 20 winter river flows exist, which cannot be explained by concomitant variations in precipitation". So you should clarify this statement somehow (as you do in the conclusion, page 17 lines 22-23), are you only referring to spring? Otherwise, it doesn't add up!**

Indeed, the sentence is confusing and refers mainly to precipitation in spring and, to a lesser extent, in fall and winter. We modified the sentence in the new version of the manuscript:

At the beginning of the Section 6:
"They are mainly linked to variations in precipitations in spring, and, to a lesser extent, in fall and winter. The question now is, therefore, to understand their origin."

**Figure corrections**
**1. Figure 1 – could you choose better colouring? the red and magenta are very similar**
Done

**2. Figure 3 – please relabel the x axes "1", "2", "3" instead with "reference", "20CRpt" and "Seine" for readability. Please also label the Y axis. The dots appear black, not blue, but their colour doesn't matter. The caption is missing an "s" on the penultimate line from "values"**
Done

**3. Figures 6 and 7 – I would relabel the y axes to say they're anomalies**
As the Figure 9 also shows anomaly, and as the Y-axe titles are already long, we left the term "anomaly" in the legend and did not add the term "anomaly" in the Y axes so it would not overload the figures.

**4. Figures 7 and 9 – this is not a colour-blind friendly palette – the Fall and Spring are not differentiable. Making the green paler and the red brighter and darker would help (see image below)**
Done

**5. Figure 8 – please label the columns and rows on the graphic itself with the variables along the top and the seasons along the right hand side to aid interpretation. I would also be tempted to reorder the rows to read spring at the top, then summer, autumn, and winter, in the order you discuss them**
Done

**6. Figure 12 – I don't think I understand this graph and its implications. The caption is VERY long. Should some of this description be in the body of the paper?**
We reduced the caption of the graph as you suggested, by moving some parts into the text. The caption is now clearer.

**7. Figure 13 – Is it sensible to reverse the scale from figs a and b? maybe use colours that don't signify "wet" and "dry"**
We modified the figure 13 by putting the same colour table in a) and b). The figure 13 is now clearer and it is easier to compare 13a) and 13b).

**8. Figure 14 – caption should read mm/d not mm/j**
Done

**9. Figure 14 – please add the crosses to the legend as 1910-1930 and 1940-1960.**
Done

**Grammar corrections:**
**Throughout – should multidecadal be hyphenated: multi-decadal?**
Both spelling seem to be valid and used in the scientific community, we kept multidecadal.

**Page 2 line 1 – "correctly reproduce", not "reproduce correctly"**
Done

**Page 2 line 4, line 19 – "river flow variability", not "flows"**
Done

**Page 2 line 11 – "for several decades", not "on"**
We changed "on" to "over", as suggested by reviewer 3.

**Page 2 line 33 – "for the second half", not "on"**
We changed "on" to "during", as suggested by reviewer 3.

**Page 4 line 1 – "this allows to study" isn't correct, I suggest "this allows studies into whether"**
Agree, we followed your suggestion.

**Page 4 line 3 – "this study therefore has" not "has therefore"**
Done

**Page 4 line 16, Page 9 line 15 – "flow" not "flows"**
Done

**Page 5 line 1, line 9 – "platform" not "plateform"**
Done

**Page 5 line 2 – what does ISBA stand for?**
ISBA stands for "Interactions between Soil, Biosphere, and Atmosphere". We added this information in the revised manuscript.

**Page 5 line 12 – "based on thousands of observation stations"**
Done

**Page 5 line 32 – im not familiar with what "weights" means**
The low-pass filter used is a weighted running average. Contrary to a classical running average in which each year within the windows receives the same weight, for the low-pass filter the years close to the window's center are given more weight than the years at the extremity.

**Page 6 line 26 – "searched for"**
We wrote "within" instead of "for"

**Page 7 – make sure your use of commas in numbers in consistent. There is 5000 and 5,000 on the same page. Find out the HESS standard**
Done

**Page 7 line 24 – recommend rewording to "this approach additionally takes local observations into account in the downscaling process..."**
Done

**Page 8 line 31 – "degrades"**
Done

**Page 8 line 32 – "developed on the Seine basin", suggest changing on to "over" or "for"**
We changed "on" to "over".

**Page 9 line 15 – "even almost as well reproduced" doesn't read well, I suggest something like "river flow variability is reproduced almost as effectively as the reference simulation"**

Agree, we rewrote it as you suggested.

**Page 9 – I don't think you need to keep re-referencing Fig 3 as often.**
Agree

**Page 9 line 28 – suggest replacing "even if" with "although"**
Agree

**Page 10 line 25 – "by a factor of 2"**
Done

**Page 10 line 27 – I think HESS prefer not to quote personal communication, as you say "might be linked", I think it's fair to leave the reference out of this sentence.**
We removed the quotation of the personal communication

**Page 11 line 18 – do you mean to reference Figure 8?**
Yes, we changed Figure 7 to Figure 8.

**Page 13 – please write NAO, AMV and SST in full, the abbreviations were last seen on page 2, a long way away!**
Done

**Page 13 line 19 – "circulation anomalies on a few decades" should possibly be "circulation anomalies for a few decades" but I don't fully understand the statement.**
We changed "on" to "for" and rephrased the sentence to make the statement more clear: "Due to its chaotic nature, the existence of significant atmospheric circulation anomalies for a few decades suggests the existence of a slow external forcing acting on the atmosphere."

**Page 14 line 13 – would we call this "paleoclimate"? It seems too modern to be considered paleo but I might be wrong**
This reconstruction is actually much longer than the period used in our study, and is a real paleoclimate reconstruction. It still covers a very long period without any direct observations and is using proxies like tree rings or ice core. Therefore, it seems quite fair to call this reconstruction a paleoclimate reconstruction of the AMV, as done by the authors of the study where this reconstruction is described.

**Page 17 line 27 – "longer" not "longest"**
Done

**Page 18 line 18 – "progress" not "progresses"**
Done

**Page 18 line 23 – suggest "climate models may have difficulties capturing some properties"**
Agree

**Response to #reviewer3:**

**Summary: First I would like to commend the authors on their rigorous response to previous comments. Their work has considerably improved the manuscript. I am now happy to recommend publication with minor corrections and do not need to see the paper again. The key points to deal with are:**

**i) some further information on the significance testing using the t-test in the results section. Some detail is in the caption of relevant figures but not in the text so far as I can see. It would help the reader if t-test was flagged in text. The word significant is used a lot in results section but no indication of when it is used in its statistical sense – please indicate using p-values throughout the manuscript.**
In the revised manuscript, we have clarified the significance tests used in the analyses.

**ii) the concluding section is long and I recommend including sub-sections to help the reader.**
We agree that this section was too long. In the revised manuscript, we extracted the discussions about limitations and assumptions from the study from the "Conclusions and perspectives" section into a new section before called "Discussion". This helped reduce the size of the section "Conclusions and perspectives".

**iii) some tightening of the language and avoidance of repetition is still needed. I indicate some points I noted below.**
We apologize for these language errors, thank you for the remarks.

**Page1**
**Line 6 – delete interesting**
Done

**Line 7 – replace reconstitution with reconstruction keep same wording thoughout. Reword to 'allows analysis of…'**
Done

**Line 12-13 – reword to 'The multidecadal hydroclimate variations also influence high and low flows …'**
Done

**Line 14-15 – reword to …'particularly through groundwater-river exchanges.'**
Done

**Line 17 – last sentence is too vague**
We rephrased the last sentence of the abstract in the new version of the manuscript: "It is therefore essential to take the associated uncertainties into account in future projections."

**Page 2**
**Line 4 – flow not flows**
Done

**Line 11 – over rather than on and delete 'especially over the coming decades'**
Done

**Line 24 – replace 'which he attributes to change' with ' attributed to changes'**
Done

**Line 32 – change to ' and North Atlantic SSTs during…'**
Done

**Page 3**
**Line 7 – delete 'by the specialists of decadal climate variability and predictability, with….and replace with , as has.**
Done

**Line 9 – delete this sentence**
Done

**Line 33 – replace sentence beginning 'A major interest…' with 'The existence of long observational series are useful for developing and evaluating the reconstructions.**
As the existence of these long-term observational series of several variables contributing to the hydrological cycle is one of the main reasons why we chose the Seine basin, we modified your sentence in order to give more importance to this point. Please find below the sentence in the revised manuscript:

"The interest of this basin is the existence of long observational series of different variables contributing to the hydrological cycle, which are useful for developing and evaluating the reconstructions."

**Page 4**
**Line 3 - delete therefore**
Done

**Line 16 – comma after series, comma after lengths**
Done

**Line 31 – not sure what daily series of reference means…delete**
"As for monthly series of reference", it is the translation of the French name of this dataset.

**Page 5**
**Line 2 – platform, not plateform – here and throughout**
Done

**Line 6 – delete the before version**
Done

**Line 21 – insert of before observation**

Done
**Line 26 – move period to before 1958-present.**
Done

**Line 27 – delete 'in the following'**
Done

**Page 6**
**Line 5 – delete 'interest of this' and 'that it is'**
Done

**Line 7 – delete beforehand**
Done

**Line 30 – replace An with the**
Done

**Page 7**
**Line 8-10 – recommend deleting this sentence as already have this information**
Done

**Line 13 – replace in with of and delete the before closest**
Done

**Line 17 sample of 5000 rather than 5000 ones**
Done

**Line 30 – replace , and with together**

Done

**Page 8**
**Line 9 – reword to…consists of selecting the Safran grid point closest to…**
Done

**Line 10 – replace on with over**
Done

**Page 9**
**Line 3 only should be after constrained**
Done

**Line 5 – replace on with for**
Done

**Line 6 – replace with with in and on with over**
Done

**Line 7 – comma after reference**
Done

**Line 8 – replace on with for**
Done

**Line 15 – delete even**
As suggested by the reviewer 2, we modified the sentence for: "At the monthly time scale, the river flow variability is reproduced almost as effectively as the reference simulation"

**Line 17 – replace rather with more**
Done

**Page 10**
**Line 7 For the later period…**
Done

**Line 12 – replace with the largest in the reconstruction**
Done

**Line 28 – his?**
It's "is", we apologize for this error.

**Page 11**

**Line 4 – 'consistent with not consistently with**
Done

**Line 5 replace on with using, delete 'As said in the introduction,'**
Done

**Line 8 – please don't call your own results interesting, just report them. Start sentence with Due to the extended length…**
Done

**Line 10 Repalce It with This**
Done

**Line 15 replace on the Seine with in the Seine**
Done

**Line 16- identified from not on**
Done

**Line 23 – what do you mean by significant is this formal test, then report test and p-value in text. Same throughout.**
We indicated the details of the significance test used at the beginning of the analysis in the new version of the manuscript:
"A Student's t-test with a p-value<0.05 is applied to evaluate the significant of changes between the negative and positive multidecadal phases."

**Line 32 flow variations not flows variations**
Done

**Page 12 Use of significant throughout needs to be more formal**
Done, see two points above

**Line 26 – insert modulating before multidecadal**
Done

**Page 13**
**Line 2 – two thirds not ⅔**
Done

**Line 5 – does this finding have implications for how hydrological models are calibrated here and in other studies?**
From a physical point of view, it is expected than soil moisture impacts the portioning of precipitation between runoff and infiltration, and hydrological models should capture that. In the physically based model that we use this partitioning is not

governed by a specific parameter. There is also no strong reason to our opinion for this impact to be different at the daily time scale or a the multidecadal time scale. But more generally it would be interesting to calibrate hydrological models during dry and wet 30-year periods and test whether their skill remains stable.

**Line 6 – replace against with over.**
Done

**Line 17 – replace lower with fewer**
Done

**Line 18 replace 'bears resemblance with the NAO, but with slight shift' with ' resembles the NAO, but with a slight..'**
Done

**Line 22 – warmer surface temperatures relative to what/when?**
We added a precision in the revised version of the manuscript:
"During the negative phases, the North Atlantic basin is characterized by warmer surface temperatures relative to the positive phases, around 0.4 to 0.7°C, especially in the subpolar gyre and the tropical North Atlantic (Figure 11b)."

**Page 14**
**Line 1 – Pacific Decadal Variability**
Done

**Line 10 – p-value?>**
We indicated the p-value in the beginning of the analysis in the revised version of the manuscript:
"As in section 5.1, the significance of changes between negative and positive multi-decadal phases of the Seine river flows are assessed with a Student's t-test with p-value<0.05."

**Line 15 – delete very interestingly**
Done

**Line 16 for the periods not on**
Done

**Line 17 – for spring river flows**
Done

**Line 18 – over the period or for the period not on the period.**
Done

**Line 21 – delete as said in the introduction**
Done

**Line 25 – on a considerably longer period**
Done

**Line 26 – connection, not teleconnection – you do not define this term.**
Done

**Page 15**
**Line 18 – they not there**
Done

**Line 20 – replace superior to with greater than**
Done

**Line 24 – reword to 'To further understand the influence…'**
Done

**Line 26 – drought in the Seine basin, not of**
Done

**Line 27 – what do you mean by strongest in reference to drought?**
We calculated a drought index in order to evaluate the strongest events in the reconstruction over the 1852-2008 period. For hydrological droughts, an index based on the VCN3 (3-day moving average minimum flow)) with a return period of 5 years is used. The severity of a drought is defined as the relative difference between the VCN3 index averaged over a given drought with the average value of the index for all droughts. We clarified this point in the new version of the manuscript:
"Droughts are defined as the periods during which the VCN3 index (3-day moving average minimum flow) remains inferior to the 5-years return value. The severity of a drought is defined as the relative difference between the VCN3 index averaged over a given drought with the average value of the index for all droughts."

**Line 30 – what is climatological groundwater levels.**
In climate, the climatological of a variable is defined as the mean state of the variable over a long period of time, typically longer than several decades.

**Page 16**
**Line 3 – delete down**

Done
**Line 4 – groundwater to river?**
Yes, corrected

**Line 8 – delete from allowed to variations.**
Done

**Line 11 – delete should it and insert it after had**
Done

**Line 19 – replace interesting with useful**
Done

**Line 17 – what do you mean by not state of the art. This is a vague statement. Please expand.**

By "not state of the art", we mean that the method used is quite old. It was developed almost twenty years ago. There have been important theoretical progresses in homogenization methods since then. Unfortunately, no new long-term homogenized precipitation and temperature series have developed in France. Modification made:
"We use monthly homogenized local precipitation and temperature observations to constrain the results of statistical downscaling in order to improve the temporal homogeneity of the reconstruction, but the homogenization method is quite old and does not benefit from recent advances in homogenization procedures."

**Page 17**
**Line 18 – Consistent with, not consistently with**
Done

**Line 20-21 – delete sentence starting Multidecadal…**
Done

**Line 23 – reword to '…until summer, regulating the runoff to precipitation ratio.'**
Done

**Line 24 – replace play with impact and delete on before river**
Done

**Line 27 – replace longest with longer**
Done

**Line 30 - replace impact of the North Pacific with connection to the North Pacific**
Done

**Line 33 – replace works with records and infirm with inform**

Done, we changed "infirm" to "deny", instead of "inform".

**Page 18**
**Line 8-9 – replace '…, that of 1921 and that of 1949' with in 1921 and 1949**
Done

**Lines 12-13 – this is a vauge sentence. Please explain how and why if retaining.**
We added a precision about that on the new version of the manuscript:

[revised manuscript text omitted]